# Taming Fat-Tailed ("Heavier-Tailed" with Potentially Infinite Variance) Noise in Federated Learning

**Haibo Yang**
Dept. of ECE
The Ohio State University
Columbus, OH 43210
yang.5952@osu.edu

**Peiwen Qiu**
Dept. of ECE
The Ohio State University
Columbus, OH 43210
qiu.617@osu.edu

**Jia Liu**
Dept. of ECE
The Ohio State University
Columbus, OH 43210
liu@ece.osu.edu

## Abstract

In recent years, federated learning (FL) has emerged as an important distributed machine learning paradigm to collaboratively learn a global model with multiple clients, while keeping data local and private. However, a key assumption in most existing works on FL algorithms' convergence analysis is that the noise in stochastic first-order information has a finite variance. Although this assumption covers all light-tailed (i.e., sub-exponential) and some heavy-tailed noise distributions (e.g., log-normal, Weibull, and some Pareto distributions), it fails for many fat-tailed noise distributions (i.e., "heavier-tailed" with potentially infinite variance) that have been empirically observed in the FL literature. To date, it remains unclear whether one can design convergent algorithms for FL systems that experience fat-tailed noise. This motivates us to fill this gap in this paper by proposing an algorithmic framework called FAT-Clipping (federated averaging with two-sided learning rates and clipping), which contains two variants: FAT-Clipping per-round (FAT-Clipping-PR) and FAT-Clipping per-iteration (FAT-Clipping-PI). Specifically, for the largest tail-index $\alpha \in (1, 2]$ such that the fat-tailed noise in FL still has a bounded $\alpha$-moment, we show that both variants achieve $\mathcal{O}((mT)^{\frac{2-\alpha}{\alpha}})$ and $\mathcal{O}((mT)^{\frac{1-\alpha}{3\alpha-2}})$ convergence rates in the strongly-convex and general non-convex settings, respectively, where $m$ and $T$ are the numbers of clients and communication rounds. Moreover, with more clipping operations compared to FAT-Clipping-PR, FAT-Clipping-PI further enjoys a linear speedup effect with respect to the number of local updates at each client and being lower-bound-matching (i.e., order-optimal). Collectively, our results advance the understanding of designing efficient algorithms for FL systems that exhibit fat-tailed first-order oracle information.

## 1 Introduction

In recent years, federated learning (FL) has emerged as an important distributed machine learning paradigm, where, coordinated by a server, a set of clients collaboratively learn a global model, while keeping their training data local and private. With intensive research in recent years, researchers have developed many FL algorithms (e.g., FedAvg [1] and many follow-ups [2–12]) that have been theoretically shown to achieve fast convergence rates in the presence of various types of randomness and heterogeneity resulted from training data, network environments, computing resources at clients, etc. Moreover, many of these algorithms enjoy the so-called "linear speedup" effect, i.e., the convergence time to a first-order stationary point is inversely proportional to the number of workers and local update steps.

However, despite the recent advances in FL algorithm design and theoretical understanding, a "cloud that remains obscures the sky of FL" is a common assumption that can be found in almost all works

36th Conference on Neural Information Processing Systems (NeurIPS 2022).

on performance analysis of FL algorithms, which states that the random noise in stochastic first-order oracles (e.g., stochastic gradients or associated estimators) has a *finite variance*. Although this assumption is not too restrictive and can cover all light-tailed (i.e., sub-exponential) and some heavy-tailed noise distributions (e.g., log-normal, Weibull, and some Pareto distributions), it fails for many "fat-tailed" distributions (i.e., "heavier-tailed" with potentially infinite variance[1]). In fact, fat-tailed distributions have already been empirically observed under centralized learning settings [15–19], let alone in the more heterogeneous FL environments. Later in Section 3, we will also provide empirical evidence that shows that fat-tailed noise distributions can be easily induced by FL systems with non-i.i.d. datasets and heterogeneous local updates across clients.

The presence of fat-tailed noise poses two major challenges in FL algorithm design and analysis: i) Experimentally, it has been shown in [20] that many existing FL algorithms suffer severely from fat-tailed noise and frequently exhibit the so-called "catastrophic failure of model performance" (i.e., sudden and dramatic drops of learning accuracy during the training phase); ii) Theoretically, the infinite variance of the random noise in the stochastic first-order oracles renders most of the proof techniques in existing FL algorithmic convergence analysis inapplicable, which necessitates new algorithmic ideas and proof strategies. In light of these empirical and theoretical challenges, two foundational questions naturally emerge in FL algorithm design and analysis: *1) Can we develop FL algorithms with convergence guarantee under fat-tailed noise? 2) If the answer to 1) is "yes," could we characterize their finite-time convergence rates?* In this paper, we provide affirmative answer to the above questions. Our major contributions in this paper are highlighted as follows:

- To address the challenges of the fat-tailed noise in FL algorithm design, we propose an algorithmic framework called FAT-Clipping (federated averaging with two-sided learning rates and clipping), which leverages a clipping technique to mitigate the impact of fat-tailed noise and uses a two-sided learning rate mechanism to lower communication complexity. Our FAT-Clipping framework contains two variants: FAT-Clipping per-round (FAT-Clipping-PR) and FAT-Clipping per-iteration (FAT-Clipping-PI). We show that, for the largest tail-index $\alpha \in (1, 2]$ such that the fat-tailed noise in FL still has a bounded $\alpha$-moment, both FAT-Clipping variants achieve $\mathcal{O}((mT)^{\frac{2-\alpha}{\alpha}})$ and $\mathcal{O}((mT)^{\frac{1-\alpha}{3\alpha-2}})$ convergence rates in the strongly-convex and general non-convex settings, respectively, where $m$ and $T$ are the numbers of clients and communication rounds.

- Between the proposed FAT-Clipping variants, FAT-Clipping-PR only performs one clipping operation in each communication round before client communicates to the server, while FAT-Clipping-PI performs clipping in each iteration of local model update. We show that, at the expense of more clipping operations compared to FAT-Clipping-PR, FAT-Clipping-PI further achieves a linear speedup effect with respect to the number local model updates at each client and is lower-bound matching in terms of convergence rate.

- In addition to theoretical analysis, we also conduct extensive numerical experiments to study the fat-tailed phenomenon in FL systems and verify the efficacy of our proposed FAT-Clipping algorithms for FL systems with fat-tailed noise. We first provide concrete empirical evidence that fail-tailed noise distributions are not uncommon in FL systems with non-i.i.d. datasets and heterogeneous local updates. We show that our FAT-Clipping algorithms render a much smoother FL training process, which effectively prevents the "catastrophic failure" in various FL settings.

For quick reference and easy comparisons, we summarize all convergence rate results in Table 1. The rest of the paper is organized as follows. In Section 2, we review the literature to put our work in comparative perspectives. In Section 3, we provide empirical fat-tailed evidence for FL to further motivate this work. Section 4 presents our FAT-Clipping algorithms and their convergence analyses. Section 5 presents numerical results and Section 6 concludes this paper. Due to space limitation, all proof details and some experiments are provided in the supplementary material.

---

[1]In the literature, the terminologies "heavy-tailed" and "fat-tailed" are not universally defined and could be interchangeable sometimes. In this paper, we follow the convention of those authors who reserve the term "fat-tailed" to mean the subclass of heavy-tailed distributions that exhibit power law decay behavior as well as infinite variance (see, e.g., [13, 14]). Thus, every fat-tailed distribution is heavy-tailed, but the reverse is not true.

Table 1: Convergence rate comparisons under fat-tailed noise distributions (shaded parts are our results; metrics: $f(\mathbf{x}) - f(\mathbf{x}^*) \leq \epsilon$ and $\|\nabla f(\mathbf{x})\| \leq \epsilon$ for strongly-convex and non-convex functions, respectively): $\alpha = 2$ and $\alpha \in (1, 2)$ correspond to non-fat-tailed and fat-tailed noises, respectively. Here, $R$ is the total number of iterations for centralized algorithms (SGD and GClip); $K$ and $T$ are local update steps and communication rounds in the FL setting, respectively; $m$ is the number of clients. N/A means no theoretical guarantee for convergence. Note that the total number of iterations $R$ in FL can be computed as $R = KT$, which relates to that in the centralized setting.

| Methods | Strongly Convex Objective Functions | | Nonconvex Objective Functions | |
| --- | --- | --- | --- | --- |
| | Fat-Tailed | Non-Fat-Tailed | Fat-Tailed | Non-Fat-Tailed |
| SGD [21] | N/A | $\mathcal{O}(R^{-1})$ | N/A | $\mathcal{O}(R^{-\frac{1}{4}})$ |
| GClip [22] | $\mathcal{O}(R^{\frac{2-2\alpha}{\alpha}})$ | $\mathcal{O}(R^{-1})$ | $\mathcal{O}(R^{\frac{1-\alpha}{3\alpha-2}})$ | $\mathcal{O}(R^{-\frac{1}{4}})$ |
| FedAvg [3, 7] | N/A | $\tilde{\mathcal{O}}((mKT)^{-1})$ | N/A | $\mathcal{O}((mKT)^{-\frac{1}{4}})$ |
| **FAT-Clipping-PR** | $\mathcal{O}((mT)^{\frac{2-2\alpha}{\alpha}}K^{\frac{2}{\alpha}})$ | $\tilde{\mathcal{O}}((mKT)^{-1})$ | $\mathcal{O}((mT)^{\frac{1-\alpha}{3\alpha-2}}K^{\frac{2-\alpha}{3\alpha-2}})$ | $\mathcal{O}((mKT)^{-\frac{1}{4}})$ |
| **FAT-Clipping-PI** | $\tilde{\mathcal{O}}((mKT)^{\frac{2-2\alpha}{\alpha}})$ | $\tilde{\mathcal{O}}((mKT)^{-1})$ | $\mathcal{O}((mKT)^{\frac{1-\alpha}{3\alpha-2}})$ | $\mathcal{O}((mKT)^{-\frac{1}{4}})$ |
| **Lower Bound** | $\Omega((mKT)^{\frac{2-2\alpha}{\alpha}})$ | $\Omega((mKT)^{-1})$ | $\Omega((mKT)^{\frac{1-\alpha}{3\alpha-2}})$ | $\Omega((mKT)^{-\frac{1}{4}})$ |

## 2 Related work

In this section, we will provide a quick overview on three related topics in the literature: i) federated learning, ii) heavy-tailed noise in learning, and iii) the clipping techniques, thus putting our work into comparative perspective to highlight our novelty and differences.

**1) Federated Learning:** As mentioned earlier, FL has recently emerged as an important distributed learning paradigm. The first and perhaps the most popular FL method, the federated averaging (FedAvg) algorithm [1], was initially proposed as a heuristic to improve communication efficiency and data privacy. Since then, FedAvg has sparked many follow-ups to further address the challenges of data/system heterogeneity and further reduce iteration and communication complexities. Notable approaches include adding regularization for the local loss function [2, 5, 6], using variance reduction techniques [3], taking adaptive learning rate strategy [8] or adaptive communication strategy [23, 24], and many momentum variants [4, 9, 10]. Empirically, these algorithms are shown to be communication-efficient [1] and enjoy better generalization performance [25]. Moreover, many state-of-the-art algorithms enjoy the "linear speedup" effect in terms of the numbers of clients and local update steps in different FL settings [3, 7, 24, 26]. We note, however, that all these theoretical results are built upon the finite variance assumption of stochastic gradient noise. Unfortunately, when the stochastic gradient noise is fat-tailed, the finite variance assumption no longer holds, and hence the associated theoretical analysis is also invalid. This motivates us to fill this gap in this paper and conduct the first theoretical analysis for FL systems that experience fat-tailed noise.

**2) Heavy-Tailed Noise in Learning:** Recently, heavy-tailed noise has been empirically observed in modern machine learning systems and theoretically analyzed [15, 16, 18, 22, 27–30]. Heavy-tailed noise significantly affects the learning dynamics and computational complexity, such as the first exit time escaping from saddle point [27] and iteration complexity [22]. This is dramatically different from classic dynamic analysis often based on sub-Gaussian noise assumption [31, 32] and algorithmic convergence analysis with bounded variance assumption [21, 33]. However, for FL, there exist few investigations about heavy-tailed behaviors. In this paper, we first demonstrate through extensive experiments that fat-tailed (i.e., heavier-tailed) noise in FL can be easily induced by data heterogeneity and local update steps. We then propose efficient algorithms to mitigate the impacts of fat-tails.

**3) The Clipping Technique:** Since our FAT-Clipping algorithms are based on the idea of clipping, here we provide an overview on this technique. As far as we know, dating back to at least 1985 [34], gradient clipping has been an effective technique to ensure convergence for optimization problems with fast-growing objective functions. In deep learning, clipping is a widely adopted technique to address the exploding gradient problem. Recently, gradient clipping was theoretically shown to be able to accelerate the training of centralized learning [17, 35–37]. Also, clipping is an effective approach to mitigate heavy-tailed noise [17, 18] in centralized learning. In FL, clipping has been used as the preconditioning step for preserving differential privacy (DP) [38–40]. Unlike these works, in this paper, we utilize clipping to address algorithmic divergence caused by fat-tailed noise in FL.

---

**Algorithm 1** Generalized FedAvg Algorithm (GFedAvg).

---

1: Initialize $\mathbf{x}_1$.
2: **for** $t = 1, \cdots, T$ (communication round) **do**
3:     **for** each client $i \in [m]$ in parallel **do**
4:         Update local model: $\mathbf{x}_{t,i}^1 = \mathbf{x}_t$.
5:         **for** $k = 1, \cdots, K$ (local update step) **do**
6:             Compute an unbiased estimate $\nabla f_i(\mathbf{x}_{t,i}^k, \xi_{t,i}^k)$ of $\nabla f_i(\mathbf{x}_{t,i}^k)$.
7:             Local update: $\mathbf{x}_{t,i}^{k+1} = \mathbf{x}_{t,i}^k - \eta_L \nabla f_i(\mathbf{x}_{t,i}^k, \xi_{t,i}^k)$.
8:         **end for**
9:         Send $\Delta_t^i = \sum_{k \in [K]} \nabla f_i(\mathbf{x}_{t,i}^k, \xi_{t,i}^k)$ to the server.
10:     **end for**
11:     Global Aggregation At Server:
12:         Receive $\Delta_t^i, i \in [m]$.
13:         Server Update: $\mathbf{x}_{t+1} = \mathbf{x}_t - \frac{\eta \eta_L}{m} \sum_{i \in [m]} \Delta_t^i$.
14:         Broadcasting $\mathbf{x}_{t+1}$ to clients.
15: **end for**

---

## 3  Fat-tailed noise phenomenon in federated learning

In this section, we first introduce the basic FL problem statement and the standard FedAvg algorithm for FL. Then, we provide some necessary background of fat-tailed distributions and provide empirical evidence to show that fat-tailed noise can be easily induced by heterogeneity of data and local updates in FL, which further motivates this work. Lastly, we demonstrate the algorithmic divergence and frequently catastrophic model failure under fat-tailed noise.

**1) Problem Statement of Federated Learning and the FedAvg Algorithm:** The goal of FL is to solve the following optimization problem:

$$\min_{\mathbf{x} \in \mathbb{R}^d} f(\mathbf{x}) := \frac{1}{m} \sum_{i=1}^m f_i(\mathbf{x}), \tag{1}$$

where $m$ is the number of clients and $f_i(\mathbf{x}) \triangleq \mathbb{E}_{\xi_i \sim D_i}[f(\mathbf{x}, \xi_i)]$ is the local loss function associated with a local data distribution $D_i$. A key challenge in FL stems from data heterogeneity, i.e., $D_i \neq D_j, \forall i \neq j$. In FL, the standard and perhaps the most popular algorithm is the federated averaging (FedAvg) method. Here in Algorithm 1, we illustrate a more generalized version of the original FedAvg (GFedAvg) with separate learning rates on the client and server sides [3,7,8]. Note that when $\eta = 1$, GFedAvg reduces to the original FedAvg [1]. In each communication round of GFedAvg, each client performs local update steps and returns the update difference $\Delta_t^i$. The server then aggregates these results and update the global model [2] and the updated model parameters will then be retrieved by the clients to start the next round of local updates.

**2) Empirical Evidence of Fat-Tailed Noise Phenomenon in Federated Learning:** With the basics of FL and the FedAvg algorithm, we are now in a position to demonstrate the empirical evidence of the existence of fat-tailed noise in FL systems. As mentioned earlier, in most performance analyses of FL algorithms, a common assumption is the bounded variance assumption of the local stochastic gradients: $\mathbb{E}[\|\nabla f_i(\mathbf{x}, \xi) - \nabla f_i(\mathbf{x})\|^2] \leq \sigma^2$. This assumption holds for all light-tailed noise distributions (i.e., the sub-exponential family) and some heavy-tailed distributions (e.g., log-normal, Weibull, and some Pareto distributions).

However, the finite-variance assumption fails to hold for many fat-tailed noise distributions. For instance, for a random variable $X$, if its density $p(x)$ has a power-law tail decreasing as $1/|x|^{\alpha+1}$ with $\alpha \in (0, 2)$, then only the $\alpha$-moment of this noise exists with $\alpha < 2$. To more precisely characterize fat-tailed distributions, in this paper, we adopt the notion of tail-index $\alpha$ [15] to parameterize fat-tailed and heavy-tailed distributions. More specifically, if the density of a random variable $X$'s distribution decays with a power law tail as $1/|x|^{\alpha+1}$ where $\alpha \in (0, 2]$, then $\alpha$ is called the *tail-infex*. This $\alpha$-parameter determines the behavior of the distribution: the smaller the $\alpha$-value, the heavier the tail

---

[2]We assume all clients participate in the training at each communication round, but the results can be extended to that with (uniformly random sampled) subset of clients in each communication round [3,7].

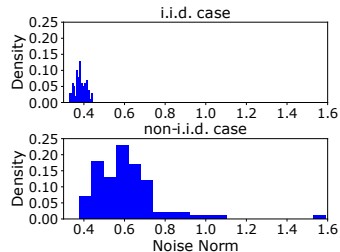
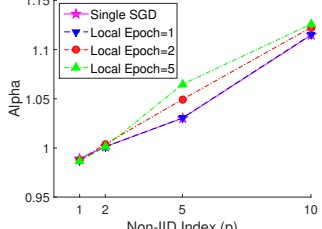
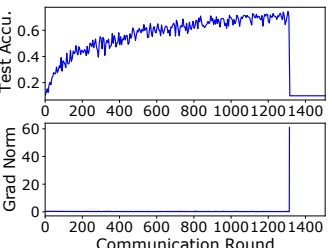

Figure 1: Distributions of the norms of the pseudo-gradient noises computed with CNN on CIFAR-10 dataset in i.i.d. case (top) and non-i.i.d. case (bottom). $m = 100$ clients participate in the training.

Figure 2: Estimation of $\alpha$ for CIFAR-10 dataset. The non-IID index $p$ represents the data heterogeneity level, and $p = 10$ is the IID case. The smaller the $p$, the more heterogeneous the data across clients.

Figure 3: Catastrophic training failures happen when applying GFedAvg on CIFAR-10 dataset, where the test accuracy experiences a sudden and dramatic drop and the pseudo-gradient norm increases substantially.

of the distribution. Also, the $\alpha$-parameter also determines the moments: $\mathbb{E}[|X|^r] < \infty$ if and only if $r < \alpha$, which implies that $X$ has infinite variance when $\alpha < 2$, i.e., being *fat-tailed*.

Next, we investigate the tail property of model updates returned by clients in the GFedAvg algorithm. Due to multiple local steps in the GFedAvg algorithm, we view the whole update vector $\Delta_t^i$ returned by each client, which we called "pseudo-gradient," as a random vector and then analyze its statistical properties. Note that in the special case with the number of local update $K = 1$, $\Delta_t^i$ coincides with a single stochastic gradient of a random sample, (i.e., $\Delta_t^i = \nabla f_i(\mathbf{x}_t, \xi_t)$).

We study the mismatch between the "non-fat-tailed" condition ($\alpha = 2$) and the empirical behavior of the stochastic psudo-gradient noise. In Fig. 1, we illustrate the distributions of the norms of the stochastic pseudo-gradient noises computed with convolutional neural network (CNN) on the CIFAR-10 dataset in both i.i.d. and non-i.i.d. client dataset settings. We can clearly observe that the non-i.i.d. case exhibits a rather fat-tailed behavior, where the pseudo-gradient norm could be as large as 1.6. Although the i.i.d. case appears to have a much lighter tail, our detailed analysis shows that it still exhibits a fat-tailed behavior. To see this, in Fig. 2, we estimate $\alpha$-value for the CIFAR-10 dataset in different scenarios: 1) different local update steps, and 2) different data heterogeneity. We use a parameter $p$ to characterize the data heterogeneity level, with $p = 10$ corresponding to the i.i.d. case. The smaller the $p$, the more heterogeneous the data among clients. Fig. 2 shows that the $\alpha$-value is smaller than 1.15 in all scenarios, and $\alpha$ increases as the non-i.i.d. index $p$ increases (i.e., closer to the i.i.d. case). This implies that the stochastic pseudo-gradient noise is fat-tailed and the "fatness" increases as the clients' data become more heterogeneous.

**3) The Impacts of Fat-Tailed Noise on Federated Learning:** Next, we show that the fat-tailed noise could lead to a "catastrophic model failure" (i.e., a sudden and dramatic drop of learning accuracy), consistent with previous observations in the FL literature [20]. To demonstrate this, we apply GFedAvg on the CIFAR-10 dataset and randomly sample five clients among $m = 10$ clients in each communication round. In Fig. 3, we illustrate a trial where a catastrophic training failure occurred. Correspondingly, we can observe in Fig. 3 a spike in the norm of the pseudo-gradient. This exceedingly large pseudo-gradient norm motivates us to apply the clipping technique to curtail the gradient updates. It is also worth noting that even if the squared norm of stochastic gradient may not be infinitely large in practice (i.e., having a bounded support empirically), it could still be too large and cause catastrophic model failures. In fact, under fat-tailed noise, the FedAvg algorithm could *diverge*, which follows from the fact that there exists one function that SGD diverges under heavy-tailed noise (see Remark 1 in [22]). As a result, the returned value by one client might be exceedingly large, leading to divergence of the FedAvg-type algorithms.

It is worth pointing out that, although we have empirically shown heavy/fat-tailed noise in FL for the first time in this paper, we are by no means the only one to have observed heavy-tailed or fat-tailed noise phenomenon property in learning. Previous works have also found heavy/fat-tailed noise phenomenon in centralized training with SGD-type algorithms. For example, the work in [15] showed the heavy-tailed noise phenomenon while (centralized) training the AlexNet on CIFAR-10. Here, we adopt a procedure similar to that in [15] to evaluate the tail index $\alpha$ of the noise norm distribution in

---

**Algorithm 2** The FAT-Clipping-PR Algorithm.

---

1: Initialize $\mathbf{x}_1$.
2: **for** $t = 1, \cdots, T$ (communication round) **do**
3:    **for** each client $i \in [m]$ in parallel **do**
4:       Update local model: $\mathbf{x}_{t,i}^1 = \mathbf{x}_t$.
5:       **for** $k = 1, \cdots, K$ (local update step) **do**
6:          Compute an unbiased estimate $\nabla f_i(\mathbf{x}_{t,i}^k, \xi_{t,i}^k)$ of $\nabla f_i(\mathbf{x}_{t,i}^k)$.
7:          Local update: $\mathbf{x}_{t,i}^{k+1} = \mathbf{x}_{t,i}^k - \eta_L \nabla f_i(\mathbf{x}_{t,i}^k, \xi_{t,i}^k)$.
8:       **end for**
9:       Let $\Delta_t^i = \sum_{k \in [K]} \nabla f_i(\mathbf{x}_{t,i}^k, \xi_{t,i}^k)$ .
10:       **Clipping:** $\tilde{\Delta}_t^i = \min\{1, \frac{\lambda}{\|\Delta_t^i\|}\}\Delta_t^i$, where $\Delta_t^i = \sum_{k \in [K]} \nabla f_i(\mathbf{x}_{t,i}^k, \xi_{t,i}^k)$.
11:       Send $\tilde{\Delta}_t^i$ to the server.
12:    **end for**
13:    Global Aggregation At Server:
14:       Receive $\tilde{\Delta}_t^i, i \in [m]$.
15:       Server Update: $\mathbf{x}_{t+1} = \mathbf{x}_t - \frac{\eta \eta_L}{m} \sum_{i \in [m]} \tilde{\Delta}_t^i$.
16:       Broadcasting $\mathbf{x}_{t+1}$ to clients.
17: **end for**

---

FL. As indicated above, we also observe that the (pseudo-)stochastic gradient noise is heavy/fat-tailed rather than Gaussian.

It is also worth noting that it remains controversial whether the heavy/fat-tailed noise phenomenon exists in all models and datasets. For example, the work in [19] showed that the stochastic gradient noise is Gaussian at least in the early phases of training, while [41] showed that the stationary distribution of stochastic gradient noise is heavy-tailed and state-dependent. Also, the evaluation methodologies of $\alpha$ could be different in different works with different statistical errors, thus leading to different observations [19, 22]. We believe that the phenomenon of heavy/fat-tailed noise in training with SGD-type methods is an under-explored area that deserves more efforts from the community.

To conclude this section, we would also like to leave a caveat regarding catastrophic training failures. In this section, we have shown that, under heavy/fat-tailed noises, catastrophic training failures happen in FL training, which is consistent with the observations in large-cohort FL training [20]. However, this does not necessarily mean that all FL trainings will suffer from catastrophic failures. Sometimes, such catastrophic failures may not happen at all (see the appendix for such empirical evidence). Here, we hypothesize that the heavy/fat-tailed noise phenomenon in FL is highly correlated with catastrophic failures in FL. This is based on our subsequent observations that such catastrophic failures in FL can be effectively mitigated by employing clipping methods. However, whether or not the heavy/fat-tailed noise phenomenon is truly the culprit for catastrophic failures still needs further investigations. Nonetheless, the mere existence of such a correlation between heavy/fat-tailed noise and catastrophic failures in FL warrants our study on mitigating heavy/fat-tailed noise in this paper.

## 4 The FAT-Clipping **algorithmic framework for fat-tailed federated learning**

Given the evidence of fat-tailed noise in FL and its potential catastrophic training failure as shown in Section 3, there is a compelling need to design an efficient FL algorithm with provable convergence guarantee under fat-tailed noise in FL. Interestingly, the observation of an exceedingly large pseudo-gradient norm in Fig. 3 suggests a natural idea to mitigate fat-tailed noise: *clipping*. Toward this end, in Section 4.1 we first propose a clipping-based algorithmic framework called FAT-Clipping, which contains two variants: FAT-Clipping per-round (FAT-Clipping-PR) and FAT-Clipping per-iteration (FAT-Clipping-PI). Then in Section 4.2, we analyze their convergence rate performances.

### 4.1 The FAT-Clipping-PR **and** FAT-Clipping-PI **algorithms**

We illustrate the FAT-Clipping-PR and FAT-Clipping-PI algorithms in Algorithms 2 and 3, respectively. It can be seen that both FAT-Clipping-PR and FAT-Clipping-PI share a similar algorithmic

---

**Algorithm 3** The FAT-Clipping-PI Algorithm.

---

1: Initialize $\mathbf{x}_1$.
2: **for** $t = 1, \cdots, T$ (communication round) **do**
3:   **for** each client $i \in [m]$ in parallel **do**
4:     Update local model: $\mathbf{x}_{t,i}^1 = \mathbf{x}_t$.
5:     **for** $k = 1, \cdots, K$ (local update step) **do**
6:       Compute an unbiased estimate $\nabla f_i(\mathbf{x}_{t,i}^k, \xi_{t,i}^k)$ of $\nabla f_i(\mathbf{x}_{t,i}^k)$.
7:       **Clipping:** $\tilde{\nabla} f_i(\mathbf{x}_{t,i}^k, \xi_{t,i}^k) = \min\{1, \frac{\lambda}{\|\nabla f_i(\mathbf{x}_{t,i}^k, \xi_{t,i}^k)\|}\} \nabla f_i(\mathbf{x}_{t,i}^k, \xi_{t,i}^k)$.
8:       Local update: $\mathbf{x}_{t,i}^{k+1} = \mathbf{x}_{t,i}^k - \eta_L \tilde{\nabla} f_i(\mathbf{x}_{t,i}^k, \xi_{t,i}^k)$.
9:     **end for**
10:      Send $\tilde{\Delta}_t^i = \sum_{k \in [K]} \tilde{\nabla} f_i(\mathbf{x}_{t,i}^k, \xi_{t,i}^k)$ to the server.
11:   **end for**
12:   Global Aggregation At Server:
13:       Receive $\tilde{\Delta}_t^i, i \in [m]$.
14:       Server Update: $\mathbf{x}_{t+1} = \mathbf{x}_t - \frac{\eta \eta_L}{m} \sum_{i \in [m]} \tilde{\Delta}_t^i$.
15:       Broadcasting $\mathbf{x}_{t+1}$ to clients.
16: **end for**

---

structure with GFedAvg, with the key differences lying in the additional clipping operations. In FAT-Clipping-PR, each client performs a clipping in each communication round on the returned $\Delta_t^i$:

$$\tilde{\Delta}_t^i = \min\left\{1, \frac{\lambda}{\|\Delta_t^i\|}\right\} \Delta_t^i, \tag{2}$$

and then sends $\tilde{\Delta}_t^i$ instead of $\Delta_t^i$ to the server (Line 10 in Algorithm 2). By contrast, in FAT-Clipping-PI, each client clips the stochastic gradient before each local update step (Line 7 in Algorithm 3):

$$\tilde{\nabla} f_i(\mathbf{x}_{t,i}^k, \xi_{t,i}^k) = \min\left\{1, \frac{\lambda}{\|\nabla f_i(\mathbf{x}_{t,i}^k, \xi_{t,i}^k)\|}\right\} \nabla f_i(\mathbf{x}_{t,i}^k, \xi_{t,i}^k), \tag{3}$$

$$\mathbf{x}_{t,i}^{k+1} = \mathbf{x}_{t,i}^k - \eta_L \tilde{\nabla} f_i(\mathbf{x}_{t,i}^k, \xi_{t,i}^k). \tag{4}$$

Then, $\tilde{\Delta}_t^i = \sum_{k \in [K]} \tilde{\nabla} f_i(\mathbf{x}_{t,i}^k, \xi_{t,i}^k)$ is sent to the server for aggregation (Line 10 in Algorithm 3).

### 4.2 Convergence analysis of the FAT-Clipping algorithms

Before conducting the convergence analysis for the FAT-Clipping algorithms, we first state two standard assumptions that are commonly used in the literature of first-order stochastic methods.

**Assumption 1** (*L*-Lipschitz Continuous Gradient). *There exists a constant $L > 0$, such that* $\|\nabla f_i(\mathbf{x}) - \nabla f_i(\mathbf{y})\| \leq L\|\mathbf{x} - \mathbf{y}\|, \forall \mathbf{x}, \mathbf{y} \in \mathbb{R}^d, and\ i \in [m]$.

**Assumption 2** (Unbiased Local Gradient Estimator). *The local gradient estimator is unbiased, i.e.,* $\mathbb{E}[\nabla f_i(\mathbf{x}, \xi)] = \nabla f_i(\mathbf{x}), \forall i \in [m]$, *where $\xi$ is a random local data sample at the $i$-th worker.*

Next, we state the key bounded $\alpha$-moment assumption for *fat-tailed* the stochastic first-order oracle, which leverages the notion of tail-index introduced in Section 3:

**Assumption 3** (Bounded $\alpha$-Moment). *There exists a real number $\alpha \in (1, 2]$ and a constant $G \geq 0$, such that $\mathbb{E}[\|\nabla f_i(\mathbf{x}, \xi)\|^\alpha] \leq G^\alpha, \forall i \in [m], \mathbf{x} \in \mathbb{R}^d$.*

**1) Convergence Rates of the FAT-Clipping-PR Algorithm:** We first state the convergence rates of FAT-Clipping-PR for $\mu$-strongly convex and non-convex objective functions.

**Theorem 1.** (Convergence Rate of FAT-Clipping-PR in the Strongly Convex Case) *Suppose that $f(\cdot)$ is a $\mu$-strongly convex function. Under Assumptions 1–3, if $\eta \eta_L K \geq \frac{2}{\mu T}$, then the output $\bar{\mathbf{x}}_T$ of FAT-Clipping-PR being chosen in such a way that $\bar{\mathbf{x}}_T = \mathbf{x}_t$ with probability $\frac{w_t}{\sum_{j \in [T]} w_j}$, where $w_t = (1 - \frac{1}{2}\mu \eta \eta_L K)^{1-t}$, satisfies:*

$$f(\bar{\mathbf{x}}_T) - f(\mathbf{x}^*) \leq \frac{\mu}{2} \exp\left(-\frac{1}{2}\mu \eta \eta_L KT\right) + \frac{\eta \eta_L K}{2} G^\alpha \lambda^{2-\alpha} + \frac{4}{\mu}\left[2G^{2\alpha}\lambda^{2-2\alpha} + 2L^2\eta_L^2 K^2 G^\alpha \lambda^{2-\alpha}\right],$$

where $\mathbf{x}^*$ denotes the global optimal solution. Further, let $\eta\eta_L K = \frac{2c}{\mu}\frac{\ln(T)}{mKT}$, where $c \geq 1$ is a constant satisfying $m^{\frac{2-2\alpha}{\alpha}} K^{\frac{2}{\alpha}} T^{c+\frac{2-2\alpha}{\alpha}} \geq 1$, and let $\eta_L \leq (mKT)^{\frac{1-\alpha}{\alpha}}$. It then follows that

$$f(\bar{\mathbf{x}}_T) - f(\mathbf{x}^*) = \mathcal{O}((mT)^{\frac{2-2\alpha}{\alpha}} K^{\frac{2}{\alpha}}).$$

**Theorem 2.** (Convergence Rate of FAT-Clipping-PR in the Nonconvex Case) *Suppose that $f(\cdot)$ is a nonconvex function. Under Assumptions 1–3, if $\eta\eta_L KL \leq 1$, then the sequence of outputs $\{\mathbf{x}_k\}$ generated by* FAT-Clipping-PR *satisfies:*

$$\min_{t\in[T]} \mathbb{E}\|\nabla f(\mathbf{x}_t)\|^2 \leq \frac{2\left(f(\mathbf{x}_1)-f(x_T)\right)}{\eta\eta_L KT} + \left(L^2\eta_L^2 K^2 G^2 + K^2 G^{2\alpha}\lambda^{-2(\alpha-1)} + L\eta_L K^2 G^{1+\alpha}\lambda^{1-\alpha}\right)$$
$$+ \frac{L\eta\eta_L}{m}\left(KG^\alpha\lambda^{2-\alpha}\right).$$

*Further, choosing learning rates and clipping parameter in such a way that $\eta\eta_L = m^{\frac{2\alpha-2}{3\alpha-2}} K^{\frac{-\alpha-2}{3\alpha-2}} T^{\frac{-\alpha}{3\alpha-2}}$, $\eta_L \leq (mT)^{\frac{1-\alpha}{3\alpha-2}} K^{\frac{4-4\alpha}{3\alpha-2}}$, and $\lambda = (mK^4 T)^{\frac{1}{3\alpha-2}}$, we have*

$$\min_{t\in[T]} \mathbb{E}\|\nabla f(\mathbf{x}_t)\|^2 = \mathcal{O}((mT)^{\frac{2-2\alpha}{3\alpha-2}} K^{\frac{4-2\alpha}{3\alpha-2}}).$$

**Remark 1.** We note that the above convergence rates for FAT-Clipping-PR does not generalize the results of FedAvg when $\alpha = 2$ (non-fat-tailed noise). Specifically, FedAvg is able to achieve $\tilde{\mathcal{O}}((mKT)^{-1})$ and $\mathcal{O}((mKT)^{-\frac{1}{4}})$ convergence rates for strongly convex ($f(\mathbf{x}) - f(\mathbf{x}^*) \leq \epsilon$) and non-convex function ($\|\nabla f(\mathbf{x})\| \leq \epsilon$), respectively [3, 42]. In contrast, FAT-Clipping-PR achieves $\mathcal{O}((mT)^{-1}K)$ and $\mathcal{O}((mT)^{-\frac{1}{4}})$ for strongly-convex and non-convex functions, respectively. These two rates are consistent with those of FedAvg in terms of $m$ and $T$, but not in terms of $K$.

Interestingly, with a separate proof for non-fat-tailed noise ($\alpha = 2$), we can show that clipping does not affect the dependence on $K$ in the convergence rates. Thus, FAT-Clipping-PR has the *same* convergence rates as those of FedAvg. Due to space limitation, we state an informal version of these theorems here. The full versions of Theorem 5 6 and their proofs are formally stated in Appendix.

**Theorem 6 & 7 (informal)** (Convergence Rates of FAT-Clipping-PR for Non-Fat-Tailed Noise): *For $\alpha = 2$, CPR-FedAvg achieves convergence rate $\tilde{\mathcal{O}}((mKT)^{-1})$ for strongly-convex and $\mathcal{O}((mKT)^{-\frac{1}{4}})$ for non-convex functions, respectively.*

**2) Convergence Rate of the FAT-Clipping-PI Algorithm:** Next, we provide the convergence rates of FAT-Clipping-PI for $\mu$-strongly convex and non-convex objective functions.

**Theorem 3.** (Convergence Rate of FAT-Clipping-PI in the Strongly Convex Case) *Suppose that $f(\cdot)$ is a $\mu$-strongly convex function. Under Assumptions 1–3, if $\eta\eta_L K \geq \frac{2}{\mu T}$, then the output $\bar{\mathbf{x}}_T$ of* FAT-Clipping-PI *being chosen in such a way that $\bar{\mathbf{x}}_T = \mathbf{x}_t$ with probability $\frac{w_t}{\sum_{j\in[T]} w_j}$, where $w_t = (1 - \frac{1}{2}\mu\eta\eta_L K)^{1-t}$, satisfies:*

$$f(\bar{\mathbf{x}}_T) - f(\mathbf{x}^*) \leq \frac{\mu}{2}\exp\left(-\frac{1}{2}\mu\eta\eta_L KT\right) + \frac{\eta\eta_L K}{2}G^\alpha\lambda^{2-\alpha}$$
$$+ \frac{4}{\mu}[2G^{2\alpha}\lambda^{-2(\alpha-1)} + 2L^2\eta_L^2 K^2 G^\alpha\lambda^{2-\alpha}],$$

*where $\mathbf{x}^*$ denotes the global optimal solution. Further, let $\eta\eta_L K = \frac{2c}{\mu}\frac{\ln(T)}{mKT}$, where $c \geq 1$ is a constant satisfying $(mK)^{\frac{2-2\alpha}{\alpha}} T^{c+\frac{2-2\alpha}{\alpha}} \geq 1$, and let $\lambda = (mKT)^{\frac{1}{\alpha}}$, and $\eta_L \leq (mT)^{-\frac{1}{2}} K^{-\frac{3}{2}}$). It then follows that*

$$f(\bar{\mathbf{x}}_T) - f(\mathbf{x}^*) = \tilde{\mathcal{O}}((mKT)^{\frac{2-2\alpha}{\alpha}}).$$

**Theorem 4.** (Convergence Rate of FAT-Clipping-PI in the Nonconvex Case) *Suppose that $f(\cdot)$ is a non-convex function. Under Assumptions 1–3, if $\eta\eta_L KL \leq 1$, then the sequence of outputs $\{\mathbf{x}_k\}$ generated by* FAT-Clipping-PI *satisfies:*

$$\min_{t\in[T]} \mathbb{E}\|\nabla f(\mathbf{x}_t)\|^2 \leq \frac{2\left(f(\mathbf{x}_1)-f(x_T)\right)}{\eta\eta_L KT} + \left(2G^{2\alpha}\lambda^{-2(\alpha-1)} + 2L^2\eta_L^2 K^2 G^\alpha\lambda^{2-\alpha}\right)$$
$$+ \frac{L\eta\eta_L}{m}\left(G^\alpha\lambda^{2-\alpha}\right).$$

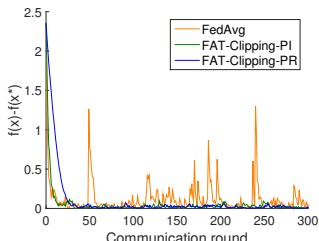 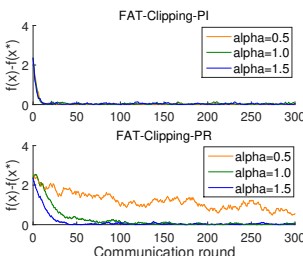 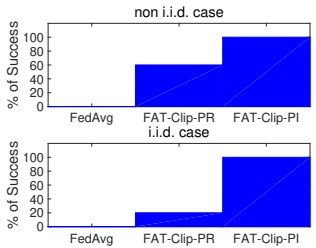

Figure 4: Convergence comparisons of FedAvg, FAT-Clipping-PI, and FAT-Clipping-PR for solving strongly convex models: synthetic data with $\xi$ having Cauchy tails (fat).

Figure 5: Convergence comparisons of FAT-Clipping-PI, and FAT-Clipping-PR for solving strongly convex models: synthetic data with $\xi$ having different fat tails represented by $\alpha$.

Figure 6: Percentage of successful training over 5 trials when applying FedAvg, FAT-Clipping-PR and FAT-Clipping-PI to CIFAR-10 dataset in non-i.i.d. case and i.i.d. case.

*Further, choosing learning rates and clipping parameter in such a way that $\eta\eta_L = m^{\frac{2\alpha-2}{3\alpha-2}}(KT)^{\frac{-\alpha}{3\alpha-2}}, \eta_L \leq (mKT)^{\frac{-\alpha}{6\alpha-4}}$, and $\lambda = (mKT)^{\frac{1}{3\alpha-2}}$, we have*

$$\min_{t\in[T]} \mathbb{E}\|\nabla f(\mathbf{x}_t)\|^2 \leq \mathcal{O}((mKT)^{\frac{2-2\alpha}{3\alpha-2}}).$$

**Remark 2.** In comparison to FAT-Clipping-PR, convergence rates of FAT-Clipping-PI generalize the results of FedAvg for the non-fat-tailed noise case (i.e., $\alpha = 2$). Specifically, when $\alpha = 2$, FAT-Clipping-PI achieves $\tilde{\mathcal{O}}((mKT)^{-1})$ and $\mathcal{O}((mKT)^{-\frac{1}{4}})$ convergence rates for strongly convex and nonconvex objective functions, respectively. These two convergence rates are consistent with those of FedAvg in terms of $m$, $K$ and $T$ (ignoring logarithmic factors in the strongly-convex case).

Next, we show that the convergence rates for FAT-Clipping-PI is *order-optimal* for $\alpha \in (1,2]$ by proving the following lower bounds.

**Corollary 1** (Convergence Rate Lower Bound). *Given any $\alpha \in (1,2]$, for any potentially randomized algorithm, there exists a stochastic strongly-convex function satisfying Assumption 3 with $G \leq 1$, such that the output of $\mathbf{x}_T$ after $T$ communication rounds has an expected error lower bounded by*

$$\mathbb{E}[f(\mathbf{x}_t)] - f(\mathbf{x}_*) = \Omega((mKT)^{\frac{2-2\alpha}{\alpha}}).$$

*Also, there exists a non-convex function satisfying Assumption 3, such that the output of $\mathbf{x}_T$ after $T$ communication rounds has an expected error lower bounded by*

$$\mathbb{E}[\|\nabla f(\mathbf{x}_t)\|]^2 = \Omega((mKT)^{\frac{2-2\alpha}{3\alpha-2}}).$$

With $T$ communication rounds, the total number of stochastic gradients is $mKT$. Thus, the lower bounds above can be obtained from the centralized SGD with fat-tailed noise [22, Theorems 5 and 6]). Clearly, the above lower bounds imply the optimality of the convergence rates of FAT-Clipping-PI .

## 5   Numerical results

In this section, we conduct numerical experiments to verify the theoretical findings in Section 4 using 1) a synthetic function, 2) a convolutional neural network (CNN) with two convolutional layers on CIFAR-10 dataset [43], and 3) RNN on Shakespeare dataset. Due to space limitation, we relegate experiment details and extra experimental results to the supplementary material.

**1) Strongly Convex Model with Synthetic Data:** We consider a strongly convex model for Problem (1) as follows: $f_i(x) = \mathbb{E}_\xi[f(x,\xi)]$ and $f(x,\xi) = \frac{1}{2}\|x\|^2 + \langle\xi, x\rangle$, where $\xi$ is a random vector. We compare FedAvg, FAT-Clipping-PI, and FAT-Clipping-PR, where the noise $\xi$ is Cauchy distributed (fat-tailed). Also, we compare FAT-Clipping-PI and FAT-Clipping-PR with $\xi$ having different tail-indexes ($\alpha = 0.5, 1.0$, and $1.5$). For each distribution, we use the same experimental setup, and $m = 5$ clients participate in the training. We show the trajectories of FedAvg, FAT-Clipping-PI, and FAT-Clipping-PR for solving Problem (1) with $\xi$ having Cauchy tails in Fig. 4 and

with $\xi$ having different $\alpha$-values in Fig. 5. We can clearly observe from Fig. 4 that FAT-Clipping-PI and FAT-Clipping-PR converge rapidly in the Cauchy case, and FAT-Clipping-PI converges faster than FAT-Clipping-PR as our theoretical results predict. In contrast, FedAvg is not convergent in the Cauchy case. In Fig. 5, we can see that the convergence processes of FAT-Clipping-PR and FAT-Clipping-PI become slower as the $\alpha$-value increases as our theoretical results predict, but the differences in FAT-Clipping-PI are much less obvious compared to those of FAT-Clipping-PR.

**2) CNN (Non-convex Model) on the CIFAR-10:** This setting has $m = 10$ clients in total, and five clients are randomly selected to participate in each round of the training. We compare FAT-Clipping algorithms with FedAvg under different data heterogeneity. To simulate data heterogeneity across clients, we distribute the data to each client in a label-based partition following the same procedure as in existing works (e.g., [1, 7, 44]): we use a parameter $p$ to represent the number of labeled classes in each client, with $p = 10$ corresponding to the i.i.d. case and the rest corresponding to non-i.i.d. cases. The smaller the $p$-value, the more heterogeneous the data across clients. In Fig. 6, we present the percentage of successful training over 5 trials when applying FedAvg, FAT-Clipping-PR and FAT-Clipping-PI on CIFAR-10 in non-i.i.d. case ($p = 2$) and i.i.d. case ($p = 10$). FAT-Clipping-PI has 100% successful rates (i.e., no catastrophic model failures) in both non-i.i.d. and i.i.d cases, and FAT-Clipping-PR has 60% and 20% successful rates in non-i.i.d. and i.i.d. cases, respectively. However, FedAvg fails in all 5 trials. Thus, compared to FedAvg, FAT-Clipping methods (FAT-Clipping-PI in particular) significantly reduce catastrophic training failures.

## 6 Conclusions and future work

In this paper, we investigated the problem of designing efficient federated learning algorithms with convergence performance guarantee in the presence of fat-tailed noise in the stochastic first-order oracles. We first showed empirical evidence that fat-tailed noise in federated learning can be induced by data heterogeneity and local update steps. To address the fat-tailed noise challenge in FL algorithm design, we proposed a clipping-based algorithmic framework called FAT-Clipping . The FAT-Clipping framework contains two variants FAT-Clipping-PR and FAT-Clipping-PI , which perform clipping operations in each communication round and in each local update step, respectively. Then, we derived the convergence rate bounds of FAT-Clipping-PR and FAT-Clipping-PI for strongly convex and non-convex loss functions under fat-tailed noise. Not only does our work shed light on theoretical understanding of FL under fat-tailed noise, it also opens the doors to many new interesting questions in FL systems that experience fat-tailed noise.

## Acknowledgements

This work has been supported in part by NSF grants CAREER CNS-2110259, CNS-2112471, ECCS-2140277, and CCF-2110252.

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
