# A Proofs for Fat-Tailed Federated Learning

## A.1 Proof of FAT-Clipping-PR

For notional clarity, we have the following update:

$$\text{Local update: } \mathbf{x}_{t,i}^{k+1} = \mathbf{x}_{t,i}^k - \eta_L \nabla f_i(\mathbf{x}_{t,i}^k, \xi_{t,i}^k), k \in [K],$$

$$\text{Clipping: } \mathbf{x}_{t,i}^{K+1} = \mathbf{x}_{t,i}^k - \eta_L \text{clipping}\left( \sum_{k \in [K]} \nabla f_i(\mathbf{x}_{t,i}^k, \xi_{t,i}^k) \right),$$

$$\Delta_{t,i} = \sum_{k \in [K]} \nabla f_i(\mathbf{x}_{t,i}^k, \xi_{t,i}^k), \tilde{\Delta}_{t,i} = \text{clipping}\left( \sum_{k \in [K]} \nabla f_i(\mathbf{x}_{t,i}^k, \xi_{t,i}^k), \lambda \right),$$

$$\Delta_t = \frac{1}{m} \sum_{i \in [m]} \Delta_{t,i}, \tilde{\Delta}_t = \frac{1}{m} \sum_{i \in [m]} \tilde{\Delta}_{t,i}$$

$$\mathbf{x}_{t+1} = \mathbf{x}_t - \eta \eta_L \tilde{\Delta}_t.$$

**Lemma 1** (Bounded Variance of Stochastic Local Updates for FAT-Clipping-PR). *Assume $f_i(\mathbf{x}, \xi)$ satisfies the Bounded $\alpha-$Moment assumption 3, then for* FAT-Clipping-PR *we have:*

$$\mathbb{E}[\|\tilde{\Delta}_t\|^2] \leq K^2 G^\alpha \lambda^{2-\alpha},$$

$$\mathbb{E}\|\tilde{\Delta}_t - \mathbb{E}[\tilde{\Delta}_t]\|^2 \leq \frac{K^2}{m} G^\alpha \lambda^{2-\alpha},$$

$$\|\frac{1}{K}\mathbb{E}[\tilde{\Delta}_t] - \nabla f(\mathbf{x}_t)\|^2 \leq L^2 \eta_L^2 K^2 G^2 + K^2 G^{2\alpha} \lambda^{-2(\alpha-1)} + L\eta_L K^2 G^{1+\alpha} \lambda^{1-\alpha}.$$

*Note here the expectation is on the random samples $\xi_{t,i}^k$.*

*Proof.*

$$\mathbb{E}[\|\tilde{\Delta}_t\|^2] = \max_{i \in [m]} \mathbb{E}[\|\tilde{\Delta}_{t,i}\|^2]$$

$$\leq \mathbb{E}[\|\tilde{\Delta}_{t,j}\|^\alpha] \lambda^{2-\alpha}$$

$$\leq K \sum_{k \in K} \mathbb{E}[\|\nabla f(\mathbf{x}_{t,j}^k, \xi_{t,j}^k)\|^\alpha] \lambda^{2-\alpha}$$

$$\leq K^2 G^\alpha \lambda^{2-\alpha},$$

where $j = argmax_{i \in [m]}\mathbb{E}[\|\tilde{\Delta}_{t,i}\|^2]$, and the first inequality is due to the clipping, i.e., $\|\tilde{\Delta}_{t,i}\| \leq \lambda$.

$$\mathbb{E}\|\tilde{\Delta}_t - \mathbb{E}[\tilde{\Delta}_t]\|^2 = \mathbb{E}\left\| \frac{1}{m} \sum_{i \in [m]} \left( \tilde{\Delta}_{t,i} - \mathbb{E}[\tilde{\Delta}_{t,i}] \right) \right\|^2$$

$$\leq \frac{1}{m^2} \sum_{i \in [m]} \mathbb{E}\|\tilde{\Delta}_{t,i} - \mathbb{E}[\tilde{\Delta}_{t,i}]\|^2$$

$$\leq \frac{1}{m^2} \sum_{i \in [m]} \mathbb{E}\|\tilde{\Delta}_{t,i}\|^2$$

$$\leq \frac{K^2}{m} G^\alpha \lambda^{2-\alpha}.$$

$$\|\frac{1}{K}\mathbb{E}[\tilde{\Delta}_t] - \nabla f(\mathbf{x}_t)\| \leq \left\| \nabla f(\mathbf{x}_t) - \frac{1}{K}\mathbb{E}[\Delta_t] \right\| + \frac{1}{K} \left\| \mathbb{E}[\Delta_t] - \mathbb{E}[\tilde{\Delta}_t] \right\|$$

$$\leq \frac{1}{mK} \sum_{i \in [m]} \sum_{k \in [K]} \mathbb{E} \left\| \nabla f_i(\mathbf{x}_t) - \nabla f_i(\mathbf{x}_{t,i}^k) \right\| + \frac{1}{mK} \sum_{i \in [m]} \left\| \mathbb{E}[\Delta_{t,i} - \tilde{\Delta}_{t,i}] \right\|$$

$$\leq \frac{L\eta_L}{mK} \sum_{i\in[m]} \sum_{k\in[K]} \mathbb{E} \left\| \sum_{j\in[k]} \nabla f_i(\mathbf{x}_{t,i}^j, \xi_{t,i}^j) \right\| + \frac{1}{mK} \sum_{i\in[m]} \mathbb{E}[\|\Delta_{t,i}\| \mathbf{1}_{\{\|\Delta_{t,i}\|\geq\lambda\}}]$$

$$\leq L\eta_L KG + KG^\alpha \lambda^{1-\alpha}$$

where $\mathbf{1}_{\{\cdot\}}$ is the indicator function, the last inequality follows from the fact that $\Delta_{t,i} = \tilde{\Delta}_{t,i}$ if $\|\Delta_{t,i}\| \leq \lambda$ and $\mathbb{E}[\|\Delta_{t,i}\| \mathbf{1}_{\{\|\Delta_{t,i}\|\geq\lambda\}}] \leq \mathbb{E}[\|\Delta_{t,i}\|^\alpha]\lambda^{1-\alpha} \leq K^2 G^\alpha \lambda^{1-\alpha}$; the second last inequality is due to L-smoothness, Jenson's inequality (i.e., $\mathbb{E}[\Delta_{t,i} - \tilde{\Delta}_{t,i}]\| \leq \mathbb{E}\|[\Delta_{t,i} - \tilde{\Delta}_{t,i}]\|$) and the clipping step. Then, we have

$$\|\frac{1}{K}\mathbb{E}[\tilde{\Delta}_t] - \nabla f(x)\|^2 \leq L^2\eta_L^2 K^2 G^2 + K^2 G^{2\alpha}\lambda^{-2(\alpha-1)} + L\eta_L K^2 G^{1+\alpha}\lambda^{1-\alpha}$$

$\square$

**Theorem 1.** (Convergence Rate of FAT-Clipping-PR in the Strongly Convex Case) *Suppose that $f(\cdot)$ is a $\mu$-strongly convex function. Under Assumptions 1–3, if $\eta\eta_L K \geq \frac{2}{\mu T}$, then the output $\bar{\mathbf{x}}_T$ of* FAT-Clipping-PR *being chosen in such a way that $\bar{\mathbf{x}}_T = \mathbf{x}_t$ with probability $\frac{w_t}{\sum_{j\in[T]} w_j}$, where $w_t = (1 - \frac{1}{2}\mu\eta\eta_L K)^{1-t}$, satisfies:*

$$f(\bar{\mathbf{x}}_T) - f(\mathbf{x}^*) \leq \frac{\mu}{2}\exp\left(-\frac{1}{2}\mu\eta\eta_L KT\right) + \frac{\eta\eta_L K}{2}G^\alpha\lambda^{2-\alpha} + \frac{4}{\mu}\left[2G^{2\alpha}\lambda^{2-2\alpha} + 2L^2\eta_L^2 K^2 G^\alpha\lambda^{2-\alpha}\right],$$

*where $\mathbf{x}^*$ denotes the global optimal solution. Further, let $\eta\eta_L K = \frac{2c}{\mu}\frac{\ln(T)}{mKT}$, where $c \geq 1$ is a constant satisfying $m^{\frac{2-2\alpha}{\alpha}} K^{\frac{2}{\alpha}} T^{c+\frac{2-2\alpha}{\alpha}} \geq 1$, and let $\eta_L \leq (mKT)^{\frac{1-\alpha}{\alpha}}$. It then follows that*

$$f(\bar{\mathbf{x}}_T) - f(\mathbf{x}^*) = \mathcal{O}((mT)^{\frac{2-2\alpha}{\alpha}} K^{\frac{2}{\alpha}}).$$

*Proof.*

$$\mathbb{E}[\|\mathbf{x}_{t+1} - x_*\|^2] = \mathbb{E}[\|\mathbf{x}_t - \eta\eta_L\tilde{\Delta}_t - x_*\|^2]$$

$$= \|\mathbf{x}_t - x_*\|^2 + \eta^2\eta_L^2\mathbb{E}[\|\tilde{\Delta}_t\|^2] - 2\left\langle \mathbf{x}_t - x_*, \eta\eta_L\left(\mathbb{E}[\tilde{\Delta}_t] - K\nabla f(\mathbf{x}_t) + K\nabla f(\mathbf{x}_t)\right)\right\rangle$$

$$\leq (1 - \mu\eta\eta_L K)\|\mathbf{x}_t - x_*\|^2 + \eta^2\eta_L^2\mathbb{E}[\|\tilde{\Delta}_t\|^2] - 2\eta\eta_L K\left\langle \mathbf{x}_t - x_*, \left(\frac{1}{K}\mathbb{E}[\tilde{\Delta}_t] - \nabla f(\mathbf{x}_t)\right)\right\rangle$$

$$\quad - 2\eta\eta_L K\left(f(\mathbf{x}_t) - f(\mathbf{x}_*)\right)$$

$$\leq (1 - \frac{1}{2}\mu\eta\eta_L K)\|\mathbf{x}_t - x_*\|^2 + \eta^2\eta_L^2\mathbb{E}[\|\tilde{\Delta}_t\|^2] + \frac{8\eta\eta_L K}{\mu}\|\frac{1}{K}\mathbb{E}[\tilde{\Delta}_t] - \nabla f(\mathbf{x}_t)\|^2$$

$$\quad - 2\eta\eta_L K\left(f(\mathbf{x}_t) - f(\mathbf{x}_*)\right).$$

The first inequality follows from the strongly-convex property, i.e., $-\langle \mathbf{x}_t - x_*, \nabla f(\mathbf{x}_t)\rangle \leq -(f(\mathbf{x}_t) - f(\mathbf{x}_*) + \frac{\mu}{2}\|x_t - \mathbf{x}_*\|^2)$, and the last inequality is due to Young's inequality. Then we have

$$f(\mathbf{x}_t) - f(\mathbf{x}_*) \leq \frac{1}{2\eta\eta_L K}\left[-\mathbb{E}[\|\mathbf{x}_{t+1} - x_*\|^2] + (1 - \frac{1}{2}\mu\eta\eta_L K)\|\mathbf{x}_t - x_*\|^2\right]$$

$$\quad + \frac{\eta\eta_L}{2K}\mathbb{E}[\|\tilde{\Delta}_t\|^2] + \frac{4}{\mu}\|\mathbb{E}[\frac{1}{K}\tilde{\Delta}_t - \nabla f(\mathbf{x}_t)\|^2$$

$$\leq \frac{1}{2\eta\eta_L K}\left[-\mathbb{E}[\|\mathbf{x}_{t+1} - x_*\|^2] + (1 - \frac{1}{2}\mu\eta\eta_L K)\|\mathbf{x}_t - x_*\|^2\right]$$

$$\quad + \frac{\eta\eta_L}{2K}K^2 G^\alpha\lambda^{2-\alpha} + \frac{4}{\mu}[L^2\eta_L^2 K^2 G^2 + K^2 G^{2\alpha}\lambda^{-2(\alpha-1)} + L\eta_L K^2 G^{1+\alpha}\lambda^{1-\alpha}],$$

where the last inequality is due to Lemma 1.

Let $w_t = (1 - \frac{1}{2}\mu\eta\eta_L K)^{1-t}$, $\bar{\mathbf{x}}_T = \mathbf{x}_t$ with probability $\frac{w_t}{\sum_{j\in[T]} w_j}$.

$$f(\bar{x}_T) - f(\mathbf{x}_*) \leq \frac{1}{\sum_{j\in[T]} w_j} \sum_{t\in[T]} \left( \frac{w_t}{2\eta\eta_L K} \left[ -\|\mathbf{x}_{t+1} - x_*\|^2 + (1 - \frac{1}{2}\mu\eta\eta_L K)\|\mathbf{x}_t - x_*\|^2 \right] \right)$$

$$+ \frac{\eta\eta_L K}{2} G^\alpha \lambda^{2-\alpha} + \frac{4}{\mu}[L^2\eta_L^2 K^2 G^2 + K^2 G^{2\alpha}\lambda^{-2(\alpha-1)} + L\eta_L K^2 G^{1+\alpha}\lambda^{1-\alpha}]$$

$$\leq \frac{1}{\sum_{j\in[T]} w_j} \sum_{t\in[T]} \left( \frac{1}{2\eta\eta_L K} \left[ -w_t\|\mathbf{x}_{t+1} - x_*\|^2 + w_{t-1}\|\mathbf{x}_t - x_*\|^2 \right] \right)$$

$$+ \frac{\eta\eta_L K}{2} G^\alpha \lambda^{2-\alpha} + \frac{4}{\mu}[L^2\eta_L^2 K^2 G^2 + K^2 G^{2\alpha}\lambda^{-2(\alpha-1)} + L\eta_L K^2 G^{1+\alpha}\lambda^{1-\alpha}]$$

$$\leq \frac{1}{\sum_{j\in[T]} w_j} \frac{1}{2\eta\eta_L K}\|\mathbf{x}_1 - x_*\|^2$$

$$+ \frac{\eta\eta_L K}{2} G^\alpha \lambda^{2-\alpha} + \frac{4}{\mu}[L^2\eta_L^2 K^2 G^2 + K^2 G^{2\alpha}\lambda^{-2(\alpha-1)} + L\eta_L K^2 G^{1+\alpha}\lambda^{1-\alpha}],$$

where the second inequality follows from $w_t \leq w_{t-1}$.

$$2\eta\eta_L K \sum_{t\in[T]} w_t = 2\eta\eta_L K \left(1 - \frac{1}{2}\mu\eta\eta_L K\right)^{-T} \sum_{t\in[T]} \left(1 - \frac{1}{2}\mu\eta\eta_L K\right)^t$$

$$= \frac{4}{\mu}\left(1 - \frac{1}{2}\mu\eta\eta_L K\right)^{-T}\left[1 - \left(1 - \frac{1}{2}\mu\eta\eta_L K\right)^T\right]$$

$$\geq \frac{4}{\mu}\left(1 - \frac{1}{2}\mu\eta\eta_L K\right)^{-T}\left[1 - \exp\left(-\frac{1}{2}\mu\eta\eta_L KT\right)\right]$$

$$\geq \frac{2}{\mu}\left(1 - \frac{1}{2}\mu\eta\eta_L K\right)^{-T},$$

where the last inequality follows from that $\eta\eta_L K \geq \frac{2}{\mu T}$, the second last inequality is due to $\left(1 - \frac{1}{2}\mu\eta\eta_L K\right)^T \leq \exp\left(-\frac{1}{2}\mu\eta\eta_L KT\right)$.

$$f(\bar{x}_T) - f(\mathbf{x}_*) \leq \frac{\mu}{2}\left(1 - \frac{1}{2}\mu\eta\eta_L K\right)^T + \frac{\eta\eta_L K}{2} G^\alpha \lambda^{2-\alpha}$$

$$+ \frac{4}{\mu}[L^2\eta_L^2 K^2 G^2 + K^2 G^{2\alpha}\lambda^{-2(\alpha-1)} + L\eta_L K^2 G^{1+\alpha}\lambda^{1-\alpha}]$$

$$\leq \frac{\mu}{2}\exp\left(-\frac{1}{2}\mu\eta\eta_L KT\right) + \frac{\eta\eta_L K}{2} G^\alpha \lambda^{2-\alpha}$$

$$+ \frac{4}{\mu}[L^2\eta_L^2 K^2 G^2 + K^2 G^{2\alpha}\lambda^{-2(\alpha-1)} + L\eta_L K^2 G^{1+\alpha}\lambda^{1-\alpha}].$$

Let $\eta\eta_L K = \frac{2c}{\mu}\frac{\ln(T)}{mKT}$ ($c \geq 1$ is a constant and $T^{-c-\frac{2-2\alpha}{\alpha}} \leq m^{\frac{2-2\alpha}{\alpha}}K^{\frac{2}{\alpha}}$), $\lambda = (mKT)^{\frac{1}{\alpha}}$, and $\eta_L \leq (mKT)^{\frac{1-\alpha}{\alpha}}$,

$$f(\bar{x}_T) - f(\mathbf{x}_*) \leq \frac{1}{T^c} + (mKT)^{\frac{2-2\alpha}{\alpha}}\ln(T) + (mT)^{\frac{2-2\alpha}{\alpha}}K^{\frac{2}{\alpha}} = \mathcal{O}((mT)^{\frac{2-2\alpha}{\alpha}}K^{\frac{2}{\alpha}}).$$

$\square$

**Theorem 2.** (Convergence Rate of FAT-Clipping-PR in the Nonconvex Case) *Suppose that $f(\cdot)$ is a nonconvex function. Under Assumptions 1–3, if $\eta\eta_L KL \leq 1$, then the sequence of outputs $\{\mathbf{x}_k\}$ generated by* FAT-Clipping-PR *satisfies:*

$$\min_{t\in[T]} \mathbb{E}\|\nabla f(\mathbf{x}_t)\|^2 \leq \frac{2(f(\mathbf{x}_1) - f(x_T))}{\eta\eta_L KT} + \left(L^2\eta_L^2 K^2 G^2 + K^2 G^{2\alpha}\lambda^{-2(\alpha-1)} + L\eta_L K^2 G^{1+\alpha}\lambda^{1-\alpha}\right)$$

$$+ \frac{L\eta\eta_L}{m}\left(KG^\alpha\lambda^{2-\alpha}\right).$$

*Further, choosing learning rates and clipping parameter in such a way that* $\eta\eta_L = m^{\frac{2\alpha-2}{3\alpha-2}}K^{\frac{-\alpha-2}{3\alpha-2}}T^{\frac{-\alpha}{3\alpha-2}}, \eta_L \le (mT)^{\frac{1-\alpha}{3\alpha-2}}K^{\frac{4-4\alpha}{3\alpha-2}},$ *and* $\lambda = (mK^4T)^{\frac{1}{3\alpha-2}},$ *we have*

$$\min_{t\in[T]}\mathbb{E}\|\nabla f(\mathbf{x}_t)\|^2 = \mathcal{O}((mT)^{\frac{2-2\alpha}{3\alpha-2}}K^{\frac{4-2\alpha}{3\alpha-2}}).$$

*Proof.* Due to the smoothness in Assumption 1, taking expectation of $f(\mathbf{x}_{t+1})$ over the randomness at communication round $t$, we have:

$$\mathbb{E}[f(\mathbf{x}_{t+1})] - f(\mathbf{x}_t) \le \left\langle \nabla f(\mathbf{x}_t), \mathbb{E}[\mathbf{x}_{t+1} - \mathbf{x}_t]\right\rangle + \frac{L}{2}\mathbb{E}[\|\mathbf{x}_{t+1} - \mathbf{x}_t\|^2]$$

$$= -\eta\eta_L\left\langle \nabla f(\mathbf{x}_t), \mathbb{E}[\tilde{\Delta}_t]\right\rangle + \frac{L}{2}\eta^2\eta_L^2\mathbb{E}[\|\tilde{\Delta}_t\|^2]$$

$$= -\frac{\eta\eta_L K}{2}\|\nabla f(\mathbf{x}_t)\|^2 - \frac{\eta\eta_L}{2K}\|\mathbb{E}[\tilde{\Delta}_t]\|^2 + \frac{\eta\eta_L K}{2}\|\nabla f(\mathbf{x}_t) - \frac{1}{K}\mathbb{E}[\tilde{\Delta}_t]\|^2 + \frac{L\eta^2\eta_L^2}{2}\mathbb{E}[\|\tilde{\Delta}_t\|^2]$$

$$= -\frac{\eta\eta_L K}{2}\|\nabla f(\mathbf{x}_t)\|^2 + \left(-\frac{\eta\eta_L}{2K} + \frac{L\eta^2\eta_L^2}{2}\right)\|\mathbb{E}[\tilde{\Delta}_t]\|^2 + \frac{\eta\eta_L K}{2}\|\nabla f(\mathbf{x}_t) - \frac{1}{K}\mathbb{E}[\tilde{\Delta}_t]\|^2$$

$$+ \frac{L\eta^2\eta_L^2}{2}\mathbb{E}[\|\tilde{\Delta}_t - \mathbb{E}[\tilde{\Delta}_t]\|^2]$$

$$\le -\frac{\eta\eta_L K}{2}\|\nabla f(\mathbf{x}_t)\|^2 + \frac{\eta\eta_L K}{2}\underbrace{\|\nabla f(\mathbf{x}_t) - \frac{1}{K}\mathbb{E}[\tilde{\Delta}_t]\|^2}_{A_1} + \frac{L\eta^2\eta_L^2}{2}\underbrace{\mathbb{E}[\|\tilde{\Delta}_t - \mathbb{E}[\tilde{\Delta}_t]\|^2]}_{A_2}, \qquad (5)$$

where the last inequality follows from $\left(-\frac{\eta\eta_L}{2K} + \frac{L\eta^2\eta_L^2}{2}\right) \le 0$ if $\eta\eta_L KL \le 1$.

From Lemma 1, we have the bound of $A_1$ and $A_2$ in (5). By rearranging and telescoping, we have:

$$\frac{1}{T}\sum_{t\in[T]}\mathbb{E}\|\nabla f(\mathbf{x}_t)\|^2 \le \frac{2\left(f(\mathbf{x}_1) - f(x_T)\right)}{\eta\eta_L KT} + \left(L^2\eta_L^2 K^2 G^2 + K^2 G^{2\alpha}\lambda^{-2(\alpha-1)} + L\eta_L K^2 G^{1+\alpha}\lambda^{1-\alpha}\right)$$

$$+ \frac{L\eta\eta_L}{m}\left(KG^\alpha\lambda^{2-\alpha}\right).$$

Suppose $\eta\eta_L = m^{\frac{2\alpha-2}{3\alpha-2}}K^{\frac{-\alpha-2}{3\alpha-2}}T^{\frac{-\alpha}{3\alpha-2}}, \eta_L \le (mT)^{\frac{1-\alpha}{3\alpha-2}}K^{\frac{4-4\alpha}{3\alpha-2}},$ and $\lambda = (mK^4T)^{\frac{1}{3\alpha-2}},$

$$\min_{t\in[T]}\mathbb{E}\|\nabla f(\mathbf{x}_t)\|^2 \le \mathcal{O}((mT)^{\frac{2-2\alpha}{3\alpha-2}}K^{\frac{4-2\alpha}{3\alpha-2}}).$$

$\square$

## A.2 Proof of FAT-Clipping-PI

For FAT-Clipping-PI, we have the following notions:

$$\tilde{\nabla}f_i(\mathbf{x}_{t,i}^k, \xi_{t,i}^k) = \min\{1, \frac{\lambda_t}{\|\nabla f_i(\mathbf{x}_{t,i}^k, \xi_{t,i}^k)\|}\}\nabla f_i(\mathbf{x}_{t,i}^k, \xi_{t,i}^k), \tilde{\nabla}f(\mathbf{x}_{t,i}^k) = \mathbb{E}[\tilde{\nabla}f_i(\mathbf{x}_{t,i}^k, \xi_{t,i}^k)];$$

Local steps: $\mathbf{x}_{t,i}^{k+1} = \mathbf{x}_{t,i}^k - \eta_L\tilde{\nabla}f_i(\mathbf{x}_{t,i}^k, \xi_{t,i}^k), k \in [K];$

$$\Delta_{t,i} = \sum_{k\in[K]}\nabla f_i(\mathbf{x}_{t,i}^k, \xi_{t,i}^k), \tilde{\Delta}_{t,i} = \sum_{k\in[K]}\tilde{\nabla}f_i(\mathbf{x}_{t,i}^k, \xi_{t,i}^k)$$

$$\Delta_t = \frac{1}{m}\sum_{i\in[m]}\Delta_{t,i}, \tilde{\Delta}_t = \frac{1}{m}\sum_{i\in[m]}\tilde{\Delta}_{t,i}$$

$$\mathbf{x}_{t+1} = \mathbf{x}_t - \eta\eta_L\frac{1}{m}\sum_{i\in[m]}\sum_{k\in[K]}\tilde{\nabla}f_i(\mathbf{x}_{t,i}^k, \xi_{t,i}^k) = \mathbf{x}_t - \eta\eta_L\tilde{\Delta}_t.$$

**Lemma 2** (Bounded Variance of Stochastic Local Updates for FAT-Clipping-PI). *Assume $f_i(\mathbf{x}, \xi)$ satisfies the Bounded $\alpha-$Moment assumption 3, then we have:*

$$\mathbb{E}[\|\tilde{\Delta}_t\|^2] \leq K^2 G^\alpha \lambda^{2-\alpha},$$

$$\mathbb{E}\|\tilde{\Delta}_t - \mathbb{E}[\tilde{\Delta}_t]\|^2 \leq \frac{K}{m} G^\alpha \lambda^{2-\alpha},$$

$$\|\frac{1}{K}\mathbb{E}[\tilde{\Delta}_t] - \nabla f(x)\|^2 \leq 2G^{2\alpha}\lambda^{-2(\alpha-1)} + 2L^2\eta_L^2 K^2 G^\alpha \lambda^{2-\alpha}.$$

*Proof.*

$$\mathbb{E}[\|\tilde{\Delta}_t\|^2] = \frac{1}{m}\sum_{i\in[m]}\mathbb{E}[\|\tilde{\Delta}_{t,i}\|^2]$$

$$\leq \frac{1}{m}\sum_{i\in[m]}\mathbb{E}[\|\sum_{j\in[K]}\tilde{\nabla}f(\mathbf{x}_{t,i}^j, \xi_{t,i}^j)\|^2]$$

$$\leq \frac{K}{m}\sum_{i\in[m]}\sum_{j\in[K]}\mathbb{E}[\|\tilde{\nabla}f(\mathbf{x}_{t,i}^j, \xi_{t,i}^j)\|^2]$$

$$\leq K^2 G^\alpha \lambda^{2-\alpha},$$

where the last inequality follows from the fact that $\mathbb{E}\|\tilde{\nabla}f_i(\mathbf{x}_{t,i}^k, \xi_{t,i}^k)\|^2 \leq \mathbb{E}\|\tilde{\nabla}f_i(\mathbf{x}_{t,i}^k, \xi_{t,i}^k)\|^\alpha \lambda^{2-\alpha} \leq G^\alpha\lambda^{2-\alpha}$ (see Lemma 9 in [22]).

$$\mathbb{E}\|\tilde{\Delta}_t - \mathbb{E}[\tilde{\Delta}_t]\|^2 = \mathbb{E}\left\|\frac{1}{m}\sum_{i\in[m]}\sum_{k\in[K]}\tilde{\nabla}f_i(\mathbf{x}_{t,i}^k, \xi_{t,i}^k) - \frac{1}{m}\sum_{i\in[m]}\sum_{k\in[K]}\tilde{\nabla}f_i(\mathbf{x}_{t,i}^k)\right\|^2$$

$$\leq \frac{1}{m^2}\sum_{i\in[m]}\sum_{k\in[K]}\mathbb{E}\|\tilde{\nabla}f_i(\mathbf{x}_{t,i}^k, \xi_{t,i}^k) - \tilde{\nabla}f_i(\mathbf{x}_{t,i}^k)\|^2$$

$$\leq \frac{1}{m^2}\sum_{i\in[m]}\sum_{k\in[K]}\mathbb{E}\|\tilde{\nabla}f_i(\mathbf{x}_{t,i}^k, \xi_{t,i}^k)\|^2$$

$$\leq \frac{K}{m}G^\alpha\lambda^{2-\alpha},$$

where the first inequality follows from the fact that $\{\tilde{\nabla}f_i(\mathbf{x}_{t,i}^k, \xi_{t,i}^k) - \tilde{\nabla}f_i(\mathbf{x}_{t,i}^k)\}$ form a martingale difference sequence (Lemma 4 in [3]), the second inequalities is due to $\mathbb{E}[\|X - \mathbb{E}[X]\|^2] \leq \mathbb{E}[\|X\|^2]$, and the third inequality follows from the fact that $\mathbb{E}\|\tilde{\nabla}f_i(\mathbf{x}_{t,i}^k, \xi_{t,i}^k)\|^2 \leq \mathbb{E}\|\tilde{\nabla}f_i(\mathbf{x}_{t,i}^k, \xi_{t,i}^k)\|^\alpha \lambda^{2-\alpha} \leq G^\alpha\lambda^{2-\alpha}$ (see Lemma 9 in [22]).

$$\|\frac{1}{K}\mathbb{E}[\tilde{\Delta}_t] - \nabla f(x)\|^2 \leq \left\|\frac{1}{mK}\sum_{i\in[m]}\sum k\in[K]\left(\tilde{\nabla}f_i(\mathbf{x}_{t,i}^k) - f_i(\mathbf{x}_t)\right)\right\|^2$$

$$\leq \frac{1}{mK}\sum_{i\in[m]}\sum_{k\in[K]}\left\|\tilde{\nabla}f_i(\mathbf{x}_{t,i}^k) - f_i(\mathbf{x}_t)\right\|^2$$

$$\leq \frac{1}{mK}\sum_{i\in[m]}\sum_{k\in[K]}\left(2\left\|\tilde{\nabla}f_i(\mathbf{x}_{t,i}^k) - \nabla f_i(\mathbf{x}_{t,i}^k)\right\|^2 + 2\left\|\nabla f_i(\mathbf{x}_{t,i}^k) - f_i(\mathbf{x}_t)\right\|^2\right)$$

$$\leq 2G^{2\alpha}\lambda^{-2(\alpha-1)} + 2L^2\frac{1}{mK}\sum_{i\in[m]}\sum_{k\in[K]}\|\mathbf{x}_{t,i}^k - \mathbf{x}_t\|^2$$

$$\leq 2G^{2\alpha}\lambda^{-2(\alpha-1)} + 2L^2\eta_L^2\frac{1}{mK}\sum_{i\in[m]}\sum_{k\in[K]}\left\|\sum_{j\in[K]}\nabla f(x_{t,i}^j, \xi_{t,i}^j)\right\|^2$$

$$\leq 2G^{2\alpha}\lambda^{-2(\alpha-1)} + 2L^2\eta_L^2 K^2 G^\alpha \lambda^{2-\alpha},$$

where the forth inequality is due to $\|\tilde{\nabla} f_i(\mathbf{x}) - \nabla f_i(\mathbf{x})\|^2 \leq G^{2\alpha}\lambda^{-2(\alpha-1)}$ (see Lemma 9 in [22]), and the last inequality follows from the fact that $\mathbb{E}\|\tilde{\nabla} f_i(\mathbf{x}_{t,i}^k, \xi_{t,i}^k)\|^2 \leq G^\alpha \lambda^{2-\alpha}$. $\qquad\square$

**Theorem 3.** (Convergence Rate of FAT-Clipping-PI in the Strongly Convex Case) *Suppose that $f(\cdot)$ is a $\mu$-strongly convex function. Under Assumptions 1–3, if $\eta\eta_L K \geq \frac{2}{\mu T}$, then the output $\bar{\mathbf{x}}_T$ of FAT-Clipping-PI being chosen in such a way that $\bar{\mathbf{x}}_T = \mathbf{x}_t$ with probability $\frac{w_t}{\sum_{j\in[T]} w_j}$, where $w_t = (1 - \frac{1}{2}\mu\eta\eta_L K)^{1-t}$, satisfies:*

$$f(\bar{\mathbf{x}}_T) - f(\mathbf{x}^*) \leq \frac{\mu}{2}\exp\left(-\frac{1}{2}\mu\eta\eta_L KT\right) + \frac{\eta\eta_L K}{2}G^\alpha\lambda^{2-\alpha}$$
$$+ \frac{4}{\mu}[2G^{2\alpha}\lambda^{-2(\alpha-1)} + 2L^2\eta_L^2 K^2 G^\alpha\lambda^{2-\alpha}],$$

*where $\mathbf{x}^*$ denotes the global optimal solution. Further, let $\eta\eta_L K = \frac{2c}{\mu}\frac{\ln(T)}{mKT}$, where $c \geq 1$ is a constant satisfying $(mK)^{\frac{2-2\alpha}{\alpha}} T^{c+\frac{2-2\alpha}{\alpha}} \geq 1$, and let $\lambda = (mKT)^{\frac{1}{\alpha}}$, and $\eta_L \leq (mT)^{-\frac{1}{2}}K^{-\frac{3}{2}}$). It then follows that*

$$f(\bar{\mathbf{x}}_T) - f(\mathbf{x}^*) = \tilde{\mathcal{O}}((mKT)^{\frac{2-2\alpha}{\alpha}}).$$

*Proof.* Similarly, we have the following one step iteration:

$$f(\mathbf{x}_t) - f(\mathbf{x}_*) \leq \frac{1}{2\eta\eta_L K}\left[-\mathbb{E}[\|\mathbf{x}_{t+1} - x_*\|^2] + (1 - \frac{1}{2}\mu\eta\eta_L K)\|\mathbf{x}_t - x_*\|^2\right]$$
$$+ \frac{\eta\eta_L}{2K}\mathbb{E}[\|\tilde{\Delta}_t\|^2] + \frac{4}{\mu}\|\mathbb{E}[\frac{1}{K}\tilde{\Delta}_t - \nabla f(\mathbf{x}_t)]\|^2$$
$$\leq \frac{1}{2\eta\eta_L K}\left[-\mathbb{E}[\|\mathbf{x}_{t+1} - x_*\|^2] + (1 - \frac{1}{2}\mu\eta\eta_L K)\|\mathbf{x}_t - x_*\|^2\right]$$
$$+ \frac{\eta\eta_L}{2K}K^2 G^\alpha\lambda^{2-\alpha} + \frac{4}{\mu}[2G^{2\alpha}\lambda^{-2(\alpha-1)} + 2L^2\eta_L^2 K^2 G^\alpha\lambda^{2-\alpha}],$$

where the last inequality is due to Lemma 2.

Let $w_t = (1 - \frac{1}{2}\mu\eta\eta_L K)^{1-t}$, $\bar{\mathbf{x}}_T = \mathbf{x}_t$ with probability $\frac{w_t}{\sum_{j\in[T]} w_j}$.

$$f(\bar{x}_T) - f(\mathbf{x}_*) \leq \frac{1}{\sum_{j\in[T]} w_j}\sum_{t\in[T]}\left(\frac{w_t}{2\eta\eta_L K}\left[-\|\mathbf{x}_{t+1} - x_*\|^2 + (1 - \frac{1}{2}\mu\eta\eta_L K)\|\mathbf{x}_t - x_*\|^2\right]\right)$$
$$+ \frac{\eta\eta_L K}{2}G^\alpha\lambda^{2-\alpha} + \frac{4}{\mu}[2G^{2\alpha}\lambda^{-2(\alpha-1)} + 2L^2\eta_L^2 K^2 G^\alpha\lambda^{2-\alpha}]$$
$$\leq \frac{1}{\sum_{j\in[T]} w_j}\sum_{t\in[T]}\left(\frac{1}{2\eta\eta_L K}\left[-w_t\|\mathbf{x}_{t+1} - x_*\|^2 + w_{t-1}\|\mathbf{x}_t - x_*\|^2\right]\right)$$
$$+ \frac{\eta\eta_L K}{2}G^\alpha\lambda^{2-\alpha} + \frac{4}{\mu}[2G^{2\alpha}\lambda^{-2(\alpha-1)} + 2L^2\eta_L^2 K^2 G^\alpha\lambda^{2-\alpha}]$$
$$\leq \frac{1}{\sum_{j\in[T]} w_j}\frac{1}{2\eta\eta_L K}\|\mathbf{x}_1 - x_*\|^2$$
$$+ \frac{\eta\eta_L K}{2}G^\alpha\lambda^{2-\alpha} + \frac{4}{\mu}[2G^{2\alpha}\lambda^{-2(\alpha-1)} + 2L^2\eta_L^2 K^2 G^\alpha\lambda^{2-\alpha}].$$

$$2\eta\eta_L K\sum_{t\in[T]} w_t = 2\eta\eta_L K\left(1 - \frac{1}{2}\mu\eta\eta_L K\right)^{-T}\sum_{t\in[T]}\left(1 - \frac{1}{2}\mu\eta\eta_L K\right)^t$$
$$= \frac{4}{\mu}\left(1 - \frac{1}{2}\mu\eta\eta_L K\right)^{-T}\left[1 - \left(1 - \frac{1}{2}\mu\eta\eta_L K\right)^T\right]$$

$$\geq \frac{4}{\mu}\left(1 - \frac{1}{2}\mu\eta\eta_L K\right)^{-T}\left[1 - \exp\left(-\frac{1}{2}\mu\eta\eta_L KT\right)\right]$$

$$\geq \frac{2}{\mu}\left(1 - \frac{1}{2}\mu\eta\eta_L K\right)^{-T},$$

where the last inequality follows from that $\eta\eta_L K \geq \frac{2}{\mu T}$, teh second last inequality is due to $\left(1 - \frac{1}{2}\mu\eta\eta_L K\right)^T \leq \exp\left(-\frac{1}{2}\mu\eta\eta_L KT\right)$.

$$f(\bar{x}_T) - f(\mathbf{x}_*) \leq \frac{\mu}{2}\left(1 - \frac{1}{2}\mu\eta\eta_L K\right)^T + \frac{\eta\eta_L K}{2}G^\alpha\lambda^{2-\alpha} + \frac{4}{\mu}[2G^{2\alpha}\lambda^{-2(\alpha-1)} + 2L^2\eta_L^2 K^2 G^\alpha\lambda^{2-\alpha}]$$

$$\leq \frac{\mu}{2}\exp\left(-\frac{1}{2}\mu\eta\eta_L KT\right) + \frac{\eta\eta_L K}{2}G^\alpha\lambda^{2-\alpha} + \frac{4}{\mu}[2G^{2\alpha}\lambda^{-2(\alpha-1)} + 2L^2\eta_L^2 K^2 G^\alpha\lambda^{2-\alpha}].$$

Let $\eta\eta_L K = \frac{2c}{\mu}\frac{\ln(T)}{mKT}$ ($c \geq 1$ is a constant and $T^{-c-\frac{2-2\alpha}{\alpha}} \leq (mK)^{\frac{2-2\alpha}{\alpha}}$), $\lambda = (mKT)^{\frac{1}{\alpha}}$, and $\eta_L \leq (mT)^{-\frac{1}{2}}K^{-\frac{3}{2}}$,

$$f(\bar{x}_T) - f(\mathbf{x}_*) \leq \frac{1}{T^c} + (mKT)^{\frac{2-2\alpha}{\alpha}}\ln(T) = \tilde{\mathcal{O}}((mKT)^{\frac{2-2\alpha}{\alpha}}).$$

$\square$

**Theorem 4.** (Convergence Rate of FAT-Clipping-PI in the Nonconvex Case) *Suppose that $f(\cdot)$ is a non-convex function. Under Assumptions 1–3, if $\eta\eta_L KL \leq 1$, then the sequence of outputs $\{\mathbf{x}_k\}$ generated by* FAT-Clipping-PI *satisfies:*

$$\min_{t\in[T]}\mathbb{E}\|\nabla f(\mathbf{x}_t)\|^2 \leq \frac{2(f(\mathbf{x}_1) - f(x_T))}{\eta\eta_L KT} + \left(2G^{2\alpha}\lambda^{-2(\alpha-1)} + 2L^2\eta_L^2 K^2 G^\alpha\lambda^{2-\alpha}\right)$$

$$+ \frac{L\eta\eta_L}{m}\left(G^\alpha\lambda^{2-\alpha}\right).$$

*Further, choosing learning rates and clipping parameter in such a way that $\eta\eta_L = m^{\frac{2\alpha-2}{3\alpha-2}}(KT)^{\frac{-\alpha}{3\alpha-2}}, \eta_L \leq (mKT)^{\frac{-\alpha}{6\alpha-4}}$, and $\lambda = (mKT)^{\frac{1}{3\alpha-2}}$, we have*

$$\min_{t\in[T]}\mathbb{E}\|\nabla f(\mathbf{x}_t)\|^2 \leq \mathcal{O}((mKT)^{\frac{2-2\alpha}{3\alpha-2}}).$$

*Proof.* Due to the smoothness in Assumption 1, taking expectation of $f(\mathbf{x}_{t+1})$ over the randomness at communication round $t$, we have:

$$\mathbb{E}[f(\mathbf{x}_{t+1})] - f(\mathbf{x}_t) \leq \langle\nabla f(\mathbf{x}_t), \mathbb{E}[\mathbf{x}_{t+1} - \mathbf{x}_t]\rangle + \frac{L}{2}\mathbb{E}[\|\mathbf{x}_{t+1} - \mathbf{x}_t\|^2]$$

$$= -\eta\eta_L\langle\nabla f(\mathbf{x}_t), \mathbb{E}[\tilde{\Delta}_t]\rangle + \frac{L}{2}\eta^2\eta_L^2\mathbb{E}[\|\tilde{\Delta}_t\|^2]$$

$$= -\frac{\eta\eta_L K}{2}\|\nabla f(\mathbf{x}_t)\|^2 - \frac{\eta\eta_L}{2K}\|\mathbb{E}[\tilde{\Delta}_t]\|^2 + \frac{\eta\eta_L K}{2}\|\nabla f(\mathbf{x}_t) - \frac{1}{K}\mathbb{E}[\tilde{\Delta}_t]\|^2 + \frac{L\eta^2\eta_L^2}{2}\mathbb{E}[\|\tilde{\Delta}_t\|^2]$$

$$= -\frac{\eta\eta_L K}{2}\|\nabla f(\mathbf{x}_t)\|^2 + \left(-\frac{\eta\eta_L}{2K} + \frac{L\eta^2\eta_L^2}{2}\right)\|\mathbb{E}[\tilde{\Delta}_t]\|^2 + \frac{\eta\eta_L K}{2}\|\nabla f(\mathbf{x}_t) - \frac{1}{K}\mathbb{E}[\tilde{\Delta}_t]\|^2 + \frac{L\eta^2\eta_L^2}{2}\mathbb{E}[\|\tilde{\Delta}_t - \mathbb{E}[\tilde{\Delta}_t]\|^2]$$

$$\leq -\frac{\eta\eta_L K}{2}\|\nabla f(\mathbf{x}_t)\|^2 + \frac{\eta\eta_L K}{2}\underbrace{\|\nabla f(\mathbf{x}_t) - \frac{1}{K}\mathbb{E}[\tilde{\Delta}_t]\|^2}_{A_1} + \frac{L\eta^2\eta_L^2}{2}\underbrace{\mathbb{E}[\|\tilde{\Delta}_t - \mathbb{E}[\tilde{\Delta}_t]\|^2]}_{A_2}, \qquad (6)$$

where the last inequality follows from $\left(-\frac{\eta\eta_L}{2K} + \frac{L\eta^2\eta_L^2}{2}\right) \leq 0$ if $\eta\eta_L KL \leq 1$.

From Lemma 2, we have the bound of $A_1$ and $A_2$ in (6). By rearranging and telescoping, we have:

$$\frac{1}{T}\sum_{t\in[T]}\mathbb{E}\|\nabla f(\mathbf{x}_t)\|^2 \leq \frac{2(f(\mathbf{x}_1) - f(x_T))}{\eta\eta_L KT} + \left(2G^{2\alpha}\lambda^{-2(\alpha-1)} + 2L^2\eta_L^2 K^2 G^\alpha\lambda^{2-\alpha}\right) + \frac{L\eta\eta_L}{m}\left(G^\alpha\lambda^{2-\alpha}\right).$$

Suppose $\eta\eta_L = m^{\frac{2\alpha-2}{3\alpha-2}}(KT)^{\frac{-\alpha}{3\alpha-2}}$, $\eta_L \leq (mKT)^{\frac{-\alpha}{6\alpha-4}}$, and $\lambda = (mKT)^{\frac{1}{3\alpha-2}}$,

$$\min_{t\in[T]} \mathbb{E}\|\nabla f(\mathbf{x}_t)\|^2 \leq \mathcal{O}((mKT)^{\frac{2-2\alpha}{3\alpha-2}}).$$

$\square$

### A.3 Proof of FAT-Clipping-PR in Gaussian Noise

In this subsection, we utilize the classic bounded variance and bounded gradient assumption.

**Assumption 4.** *(Bounded Stochastic Gradient Variance) There exists a constant $\sigma > 0$, such that the variance of each local gradient estimator is bounded by $\mathbb{E}[\|\nabla f_i(\mathbf{x},\xi) - \nabla f_i(\mathbf{x})\|^2] \leq \sigma^2$, $\forall i \in [m]$.*

**Assumption 5.** *(Bounded Gradient ) There exists a constant $G \geq 0$, such that gradient is bounded by $\|\nabla f_i(\mathbf{x})\|^2 \leq G^2$, $\forall i \in [m]$.*

**Lemma 3** (Lemma F.5 [18])**.** *Suppose there exists a constant $\sigma$ such that the variance of the stochastic gradient of $F$ has bounded variance, i.e., $\mathbb{E}[\|\nabla F(\mathbf{x},\xi) - \nabla F(\mathbf{x})\|^2] \leq \sigma^2$, and $\|\nabla F(\mathbf{x})\|^2 \leq \frac{\lambda}{2}$, then we have the following inequalities for the clipping $\tilde{\nabla}F(\mathbf{x}_t) = \mathbb{E}[\tilde{\nabla}F(\mathbf{x},\xi)] = \mathbb{E}[\min\{1, \frac{\lambda}{\|\nabla F(\mathbf{x},\xi)\|}\}\nabla F(\mathbf{x},\xi)]$:*

$$\|\mathbb{E}[\tilde{\nabla}F(\mathbf{x},\xi)] - \nabla F(\mathbf{x})\|^2 \leq \frac{16\sigma^4}{\lambda^2},$$

$$\mathbb{E}\|\tilde{\nabla}F(\mathbf{x},\xi) - \nabla F(\mathbf{x})\|^2 \leq 18\sigma^2,$$

$$\mathbb{E}\|\tilde{\nabla}F(\mathbf{x},\xi) - \mathbb{E}[\tilde{\nabla}F(\mathbf{x},\xi)]\|^2 \leq 18\sigma^2.$$

We remark that for any stochastic estimator satisfies the above conditions, the above inequalities hold. The proof is the exactly same as that in original proof [18].

**Lemma 4** (Bounded Variance of Clipping Stochastic Local Updates in FAT-Clipping-PR)**.** *Assume $f_i$ satisfies the bounded variance assumption, then we have:*

$$\mathbb{E}[\|\Delta_{t,i} - \mathbb{E}[\Delta_{t,i}]\|^2] \leq K\sigma^2.$$

*In addition, assume there exists a constant $G$ such that gradient is bounded $\|\nabla f_i(\mathbf{x})\|^2 \leq G^2$, if we set clipping parameter as $\lambda^2 \geq 2K^2G^2$, i.e., $\|\nabla f_i(\mathbf{x})\| \leq \frac{\lambda}{2}$, then we have:*

$$\left\|\mathbb{E}[\Delta_{t,i}] - \mathbb{E}[\tilde{\Delta}_{t,i}]\right\|^2 \leq \frac{16K\sigma^4}{\lambda^2},$$

$$\mathbb{E}\|\tilde{\Delta}_t - \mathbb{E}[\tilde{\Delta}_t]\|^2 \leq \frac{18K}{m}\sigma^4.$$

.

*Proof.*

$$\mathbb{E}[\|\Delta_{t,i} - \mathbb{E}[\Delta_{t,i}]\|^2] = \mathbb{E}[\|\nabla f(\mathbf{x}_{t,i}^j,\xi_{t,i}^j) - \mathbb{E}[\nabla f(\mathbf{x}_{t,i}^j)]\|^2]$$
$$\leq K\sigma^2,$$

where $\{\nabla f(\mathbf{x}_{t,i}^j,\xi_{t,i}^j) - \mathbb{E}[\nabla f(\mathbf{x}_{t,i}^j)]\}$ forms martingale difference sequence (Lemma 4 in [3]).

Then by applying Lemma 3, we have the bound of $\left\|\mathbb{E}[\Delta_{t,i}] - \mathbb{E}[\tilde{\Delta}_{t,i}]\right\|^2$.

$$\mathbb{E}\|\tilde{\Delta}_t - \mathbb{E}[\tilde{\Delta}_t]\|^2 = \mathbb{E}\left\|\frac{1}{m}\sum_{i\in[m]}\tilde{\Delta}_{t,i} - \frac{1}{m}\sum_{i\in[m]}\mathbb{E}[\tilde{\Delta}_{t,i}]\right\|^2$$
$$\leq \frac{18K}{m}\sigma^4.$$

where the last inequality follows from the fact that $\mathbb{E}[\|\Delta_{t,i} - \mathbb{E}[\Delta_{t,i}]\|^2] \leq K\sigma^2$, $\{\Delta_{t,i} - \mathbb{E}[\Delta_{t,i}]\}$ forms martingale difference sequence and Lemma 3. $\square$

**Theorem 5.** *Suppose $f$ is non-convex function, under Assumptions 1, 2, 4, and 5, if $\eta\eta_L K L \leq 1$, then the sequence of outputs $\{\mathbf{x}_k\}$ generated by Algorithm* FAT-Clipping-PR *satisfies:*

$$\frac{1}{T}\sum_{t\in[T]}\mathbb{E}\|\nabla f(\mathbf{x}_t)\|^2 \leq \frac{2\left(f(\mathbf{x}_0)-f(\mathbf{x}_T)\right)}{\eta\eta_L KT} + \frac{1}{T}\sum_{t\in[T]}\left(2L^2\eta_L^2 K^2(\sigma^2+G^2)+\frac{32\sigma^4}{K^2\lambda_t^2}\right)+\left(\frac{18L\eta\eta_L}{m}\sigma^2\right).$$

*Choosing learning rates and clipping parameter as $\eta\eta_L = \frac{m^{1/2}}{(KT)^{1/2}}, \eta_L \leq \frac{1}{(mT)^{1/2}K^{5/2}}$, and $\lambda_t \geq (mT)^{1/4}K^{-3/4}$,*

$$\min_{t\in[T]}\mathbb{E}\|\nabla f(\mathbf{x}_t)\|^2 \leq \mathcal{O}((mKT)^{-\frac{1}{2}}).$$

*Proof.* Due to the smoothness in Assumption 1, taking expectation of $f(\mathbf{x}_{t+1})$ over the randomness at communication round $t$, we have the same inequality:

$$\mathbb{E}[f(\mathbf{x}_{t+1})]-f(\mathbf{x}_t) \leq -\frac{\eta\eta_L K}{2}\|\nabla f(\mathbf{x}_t)\|^2 + \frac{\eta\eta_L K}{2}\underbrace{\left\|\nabla f(\mathbf{x}_t)-\frac{1}{K}\mathbb{E}[\tilde{\Delta}_t]\right\|^2}_{A_1} + \frac{L\eta^2\eta_L^2}{2}\underbrace{\mathbb{E}[\|\tilde{\Delta}_t-\mathbb{E}[\tilde{\Delta}_t]\|^2]}_{A_2},$$

(7)

where it requires $\eta\eta_L K L \leq 1$.

Note that the term $A_1$ in (7) can be bounded as follows:

$$A_1 = \left\|\nabla f(\mathbf{x}_t)-\frac{1}{K}\mathbb{E}[\tilde{\Delta}_t]\right\|^2$$

$$= 2\left\|\nabla f(\mathbf{x}_t)-\frac{1}{K}\mathbb{E}[\Delta_t]\right\|^2 + \frac{2}{K^2}\left\|\mathbb{E}[\Delta_t]-\mathbb{E}[\tilde{\Delta}_t]\right\|^2$$

$$\leq \frac{2}{mK}\sum_{i\in[m]}\sum_{k\in[K]}\left\|\nabla f_i(\mathbf{x}_t)-\nabla f_i(\mathbf{x}_{t,i}^k)\right\|^2 + \frac{2}{mK^2}\sum_{i\in[m]}\left\|\mathbb{E}[\Delta_{t,i}]-\mathbb{E}[\tilde{\Delta}_{t,i}]\right\|^2$$

$$\leq \frac{2L^2\eta_L^2}{mK}\sum_{i\in[m]}\sum_{k\in[K]}\mathbb{E}\left\|\sum_{j\in[k]}\nabla f_i(\mathbf{x}_{t,i}^j,\xi_{t,i}^j)\right\|^2 + \frac{2}{mK^2}\sum_{i\in[m]}\left\|\mathbb{E}[\Delta_{t,i}]-\mathbb{E}[\tilde{\Delta}_{t,i}]\right\|^2$$

$$\leq 2L^2\eta_L^2 K^2\left(\mathbb{E}\left\|\nabla f_i(\mathbf{x}_{t,i}^k,\xi_{t,i}^k)-\nabla f_i(\mathbf{x}_{t,i}^k)\right\|^2 + \left\|\nabla f_i(\mathbf{x}_{t,i}^k)\right\|^2\right) + \frac{32\sigma^4}{K^2\lambda_t^2}$$

$$\leq 2L^2\eta_L^2 K^2(\sigma^2+G^2) + \frac{32\sigma^4}{K^2\lambda_t^2}$$

where the second inequality is due to smoothness assumption 1, the third inequality is due to Lemma 4, and the last inequality follows from bounded variance assumption 4 and bounded gradient assumption 5.

From Lemma 4, the term $A_2$ in (7) can be bounded as follows:

$$A_2 \leq \frac{18K\sigma^2}{m}.$$

Putting pieces together, we can have the one communication round descent in expectation:

$$\mathbb{E}[f(\mathbf{x}_{t+1})]-f(\mathbf{x}_t) \leq -\frac{\eta\eta_L K}{2}\|\nabla f(\mathbf{x}_t)\|^2 + \frac{\eta\eta_L K}{2}\underbrace{\left\|\nabla f(\mathbf{x}_t)-\frac{1}{K}\mathbb{E}[\tilde{\Delta}_t]\right\|^2}_{A_1} + \frac{L\eta^2\eta_L^2}{2}\underbrace{\mathbb{E}[\|\tilde{\Delta}_t-\mathbb{E}[\tilde{\Delta}_t]\|^2]}_{A_2}$$

$$\leq -\frac{\eta\eta_L K}{2}\|\nabla f(\mathbf{x}_t)\|^2 + \frac{\eta\eta_L K}{2}\left(2L^2\eta_L^2 K^2(\sigma^2+G^2)+\frac{32\sigma^4}{K^2\lambda_t^2}\right) + \frac{18LK\eta^2\eta_L^2}{2m}\sigma^2.$$

Rearranging and telescoping, we have the final convergence result:

$$\frac{1}{T}\sum_{t\in[T]}\mathbb{E}\|\nabla f(\mathbf{x}_t)\|^2 \leq \frac{2\left(f(\mathbf{x}_0)-f(\mathbf{x}_T)\right)}{\eta\eta_L KT} + \frac{1}{T}\sum_{t\in[T]}\left(2L^2\eta_L^2 K^2(\sigma^2+G^2)+\frac{32\sigma^2}{K^2\lambda_t^2}\right)+\left(\frac{18L\eta\eta_L}{m}\sigma^2\right).$$

Suppose $\eta\eta_L = \frac{m^{1/2}}{(KT)^{1/2}}, \eta_L \leq \frac{1}{(mT)^{1/2}K^{5/2}}$, and $\lambda_t \geq (mT)^{1/4}K^{-3/4}$,

$$\min_{t\in[T]} \mathbb{E}\|\nabla f(\mathbf{x}_t)\|^2 \leq \mathcal{O}((mKT)^{-\frac{1}{2}}).$$

$\square$

**Theorem 6.** *Suppose $f$ is $\mu$-strongly convex function, under Assumptions 1–3, if $\eta\eta_L K \geq \frac{2}{\mu T}$, then the outputs $\bar{\mathbf{x}}_T$ in Algorithm 2 (FAT-Clipping-PR) by $\bar{\mathbf{x}}_T = \mathbf{x}_t$ with probability $\frac{w_t}{\sum_{j\in[T]} w_j}$ where $w_t = (1 - \frac{1}{2}\mu\eta\eta_L K)^{1-t}$ satisfies:*

$$f(\bar{x}_T) - f(\mathbf{x}_*) \leq \frac{\mu}{2}\exp\left(-\frac{1}{2}\mu\eta\eta_L KT\right) + \frac{\eta\eta_L K}{2}G^2 + \frac{4}{\mu}[2L^2\eta_L^2 K^2(G^2 + \sigma^2) + \frac{32\sigma^4}{\lambda^2}].$$

*Suppose $\eta\eta_L K = \frac{2c}{\mu}\frac{\ln(T)}{mKT}$ ($c > 0$ is a constant and $T^{-c+1} \leq (mK)^{-1}$), $\lambda \geq (mKT)^{\frac{1}{2}}$, and $\eta_L \leq (mT)^{-\frac{1}{2}}K^{-\frac{3}{2}}$,*

$$f(\bar{x}_T) - f(\mathbf{x}_*) = \tilde{\mathcal{O}}((mKT)^{-1}).$$

*Proof.*

$$\|\frac{1}{K}\mathbb{E}[\tilde{\Delta}_t] - \nabla f(x)\|^2 \leq 2\left\|\nabla f(\mathbf{x}_t) - \frac{1}{K}\mathbb{E}[\Delta_t]\right\|^2 + \frac{2}{K}\left\|\mathbb{E}[\Delta_t] - \mathbb{E}[\tilde{\Delta}_t]\right\|^2$$

$$\leq \frac{2}{mK}\sum_{i\in[m]}\sum_{k\in[K]}\mathbb{E}\left\|\nabla f_i(\mathbf{x}_t) - \nabla f_i(\mathbf{x}_{t,i}^k)\right\|^2 + \frac{2}{mK}\sum_{i\in[m]}\left\|\mathbb{E}[\Delta_{t,i}] - \mathbb{E}[\tilde{\Delta}_{t,i}]\right\|^2$$

$$\leq \frac{2L^2\eta_L^2}{mK}\sum_{i\in[m]}\sum_{k\in[K]}\mathbb{E}\left\|\sum_{j\in[k]}\nabla f_i(\mathbf{x}_{t,i}^j, \xi_{t,i}^j)\right\|^2 + \frac{32\sigma^4}{\lambda^2}$$

$$\leq 2L^2\eta_L^2 K^2(G^2 + \sigma^2) + \frac{32\sigma^4}{\lambda^2}.$$

Similarly, we have

$$f(\mathbf{x}_t) - f(\mathbf{x}_*) \leq \frac{1}{2\eta\eta_L K}\left[-\mathbb{E}[\|\mathbf{x}_{t+1} - x_*\|^2] + (1 - \frac{1}{2}\mu\eta\eta_L K)\|\mathbf{x}_t - x_*\|^2\right]$$

$$+ \frac{\eta\eta_L}{2K}\mathbb{E}[\|\tilde{\Delta}_t\|^2] + \frac{4}{\mu}\|\mathbb{E}[\frac{1}{K}\tilde{\Delta}_t - \nabla f(\mathbf{x}_t)\|^2$$

$$\leq \frac{1}{2\eta\eta_L K}\left[-\mathbb{E}[\|\mathbf{x}_{t+1} - x_*\|^2] + (1 - \frac{1}{2}\mu\eta\eta_L K)\|\mathbf{x}_t - x_*\|^2\right]$$

$$+ \frac{\eta\eta_L K}{2}G^2 + \frac{4}{\mu}[2L^2\eta_L^2 K^2(G^2 + \sigma^2) + \frac{32\sigma^4}{\lambda^2}],$$

Let $w_t = (1 - \frac{1}{2}\mu\eta\eta_L K)^{1-t}$, $\bar{\mathbf{x}}_T = \mathbf{x}_t$ with probability $\frac{w_t}{\sum_{j\in[T]} w_j}$.

$$f(\bar{x}_T) - f(\mathbf{x}_*) \leq \frac{1}{\sum_{j\in[T]} w_j}\sum_{t\in[T]}\left(\frac{w_t}{2\eta\eta_L K}\left[-\|\mathbf{x}_{t+1} - x_*\|^2 + (1 - \frac{1}{2}\mu\eta\eta_L K)\|\mathbf{x}_t - x_*\|^2\right]\right)$$

$$+ \frac{\eta\eta_L K}{2}G^2 + \frac{4}{\mu}[2L^2\eta_L^2 K^2(G^2 + \sigma^2) + \frac{32\sigma^4}{\lambda^2}]$$

$$\leq \frac{1}{\sum_{j\in[T]} w_j}\frac{1}{2\eta\eta_L K}\|\mathbf{x}_1 - x_*\|^2 + \frac{\eta\eta_L K}{2}G^2 + \frac{4}{\mu}[2L^2\eta_L^2 K^2(G^2 + \sigma^2) + \frac{32\sigma^4}{\lambda^2}].$$

Same as that in heavy-tailed noise case, we have the same bound for $2\eta\eta_L K \sum_{t\in[T]} w_t$:

$$2\eta\eta_L K \sum_{t\in[T]} w_t \geq \frac{2}{\mu}\left(1 - \frac{1}{2}\mu\eta\eta_L K\right)^{-T},$$

where it requires $\eta\eta_L K \geq \frac{2}{\mu T}$.

$$f(\bar{x}_T) - f(\mathbf{x}_*) \leq \frac{\mu}{2}\left(1 - \frac{1}{2}\mu\eta\eta_L K\right)^T + \frac{\eta\eta_L K}{2}G^2 + \frac{4}{\mu}[2L^2\eta_L^2 K^2(G^2 + \sigma^2) + \frac{32\sigma^4}{\lambda^2}]$$

$$\leq \frac{\mu}{2}\exp\left(-\frac{1}{2}\mu\eta\eta_L KT\right) + \frac{\eta\eta_L K}{2}G^2 + \frac{4}{\mu}[2L^2\eta_L^2 K^2(G^2 + \sigma^2) + \frac{32\sigma^4}{\lambda^2}].$$

Let $\eta\eta_L K = \frac{2c}{\mu}\frac{\ln(T)}{mKT}$ ($c > 0$ is a constant and $T^{-c+1} \leq (mK)^{-1}$), $\lambda \geq (mKT)^{\frac{1}{2}}$, and $\eta_L \leq (mT)^{-\frac{1}{2}}K^{-\frac{3}{2}}$,

$$f(\bar{x}_T) - f(\mathbf{x}_*) = \tilde{\mathcal{O}}((mKT)^{-1}).$$

□

# B   Experiments in Section 3

In this section, we provide experimental details to demonstrate the fat-tailed noise phenomenon in federated learning. We conduct experiments with CNN on CIFAR-10 dataset as shown in Section 3, and provide additional results of RNN model on Shakespeare dataset. Furthermore, we verify the accuracy of $\alpha$ estimation with logistic regression on MNIST dataset.

## B.1   CNN on CIFAR-10 Dataset

### B.1.1   Experiment details

We run a convolutional neural network (CNN) model on CIFAR-10 dataset using FedAvg. The CNN architecture is shown in Table 2. To simulate data heterogeneity across clients, we manually distribute the the data to each client in a label-based partition. Specifically, we split the data according to the classes ($p$) of images that each client has. Then, we randomly distribute these partitioned data to $m = 100$ clients such that each client has only $p$ classes of images in both training and test data, which causes the heterogeneity of data among different clients. For example, for $p = 10$, each client contains training/test data samples with ten classes. Since CIFAR-10 has 10 classes of images, $p = 10$ is the nearly i.i.d case. For the remaining $p$, each client contains data samples with class $p$. Therefore, the classes ($p$) of images in each client's local dataset can be used to represent the non-i.i.d. degree. The smaller the $p$-value, the more heterogeneous the data between clients.

In this experimental setting, we use the global learning rate $\frac{\eta\eta_L}{m} = 1.0$ and the local learning rate $\eta_L = 0.1$. The batch size is set to 500, and the communication round is $T = 4000$. We run this experiment in different cases, including singleSGD and different local epochs $\{1, 2, 5\}$ and non-iid index $p \in \{1, 2, 5, 10\}$. Single SGD means one local update step, which is equivalent to mini-batch SGD.

Table 2: CNN architecture for CIFAR-10.

| LAYER TYPE | SIZE |
|---|---|
| Convolution + ReLu | $3 \times 32 \times 5$ |
| Max Pooling | $2 \times 2$ |
| Convolution + ReLu | $32 \times 64 \times 5$ |
| Max Pooling | $2 \times 2$ |
| Fully Connected + ReLU | $1600 \times 512$ |
| Fully Connected + ReLU | $512 \times 128$ |
| Fully Connected | $128 \times 10$ |

### B.1.2 Additional experimental results

We provide additional distributions of the norms of the pseudo-gradient noises in different cases as follows. From Fig. 7- 10, the observation is that the gradient norm statistics are contracted together for more iid cases while dispersed uniformly for more non-iid cases. This is

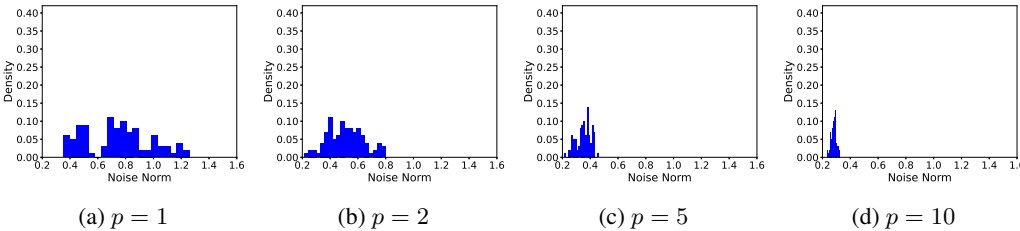

(a) $p = 1$     (b) $p = 2$     (c) $p = 5$     (d) $p = 10$

Figure 7: Distributions of the norms of the pseudo-gradient noises for CIFAR-10 dataset in the case of *Single SGD*.

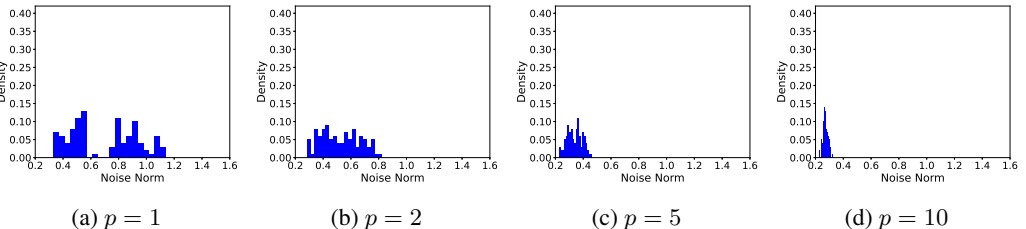

(a) $p = 1$     (b) $p = 2$     (c) $p = 5$     (d) $p = 10$

Figure 8: Distributions of the norms of the pseudo-gradient noises for CIFAR-10 dataset in the case of *Local Epoch=1*.

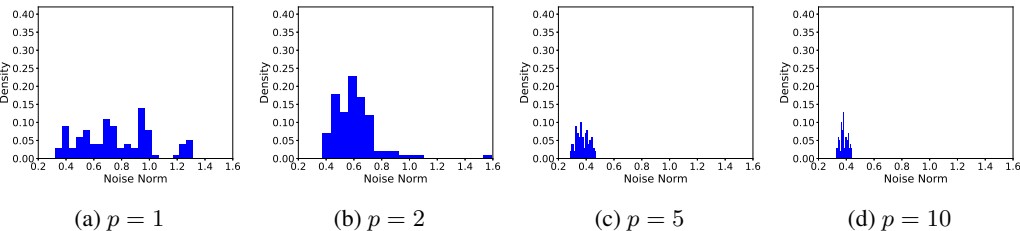

(a) $p = 1$  (b) $p = 2$  (c) $p = 5$  (d) $p = 10$

Figure 9: Distributions of the norms of the pseudo-gradient noises for CIFAR-10 dataset in the case of *Local Epoch=2*.

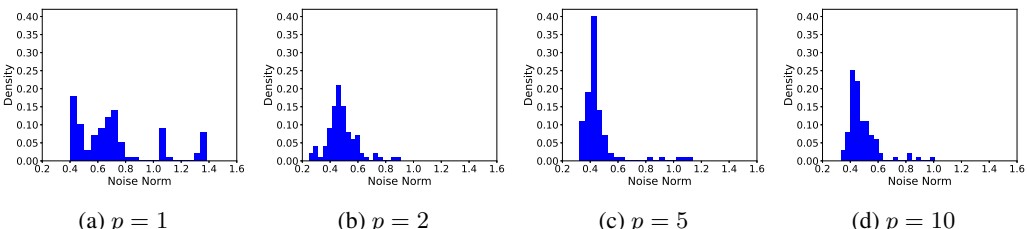

(a) $p = 1$  (b) $p = 2$  (c) $p = 5$  (d) $p = 10$

Figure 10: Distributions of the norms of the pseudo-gradient noises for CIFAR-10 dataset in the case of *Local Epoch=5*.

## B.2   RNN on Shakespeare Dataset

### B.2.1   Experiment details

To provide more evidences of the fat-tailed noise phenomenon, we further run a recurrent neural network (RNN) model on Shakespeare dataset.

Shakespeare dataset is a natural non-iid dataset, and it is built from *The Complete Works of William Shakespeare* [1]. The learning task is to predict next character, and there are 80 classes of characters in total. We use a two-layer LSTM classifier containing 100 hidden units with an 8-dimensional (8D) embedding layer. The model inputs a sequence of 80 characters, embeds each of the characters into a learned 8D space, and then outputs one character per training sample after two LSTM layers and a densely-connected layer. The dataset and model are taken from [45].

There are $m = 143$ clients participating in this experiment. The global learning rate is chosen as $1.0$, and the local learning rate is chosen as $0.8$. The batch size is set to $10$, and the communication round is $T = 150$.

### B.2.2   Experimental results

We show the results when local step is set to be one (Single SGD), and multiple local epochs $\{1, 2, 5\}$. In Fig. 11, we observe that the $\alpha$-value is smaller than 2, and it increases when the number of local epoch increases. This implies that the gradient noise is fat-tailed. Fig. 12 shows that the distributions of the norms of the pseudo-gradient noises are fat-tailed.

## B.3   Accuracy of Alpha Estimation (Logistic Regression on MNIST Dataset)

Accurate $\alpha$-value computation requires the full-gradient calculation, and we have to compute both full-gradient and stochastic gradient in each local step. This is computationally expensive. Instead, we use an estimation to approximate the exact $\alpha$-value. The full-gradient is replaced by the mean value of the stochastic gradients. We verify the accuracy of this estimation method by running logistic regression on MNIST dataset [46]. The details and the results are described as follows.

### B.3.1   Experiment details

MNIST dataset contains ten classes of images, and it is manually partitioned using the same method as to partition CIFAR-10 dataset (see details in Appendix B.1.1). The number of classes ($p$) that each client has can be used to represent the non-iid level.

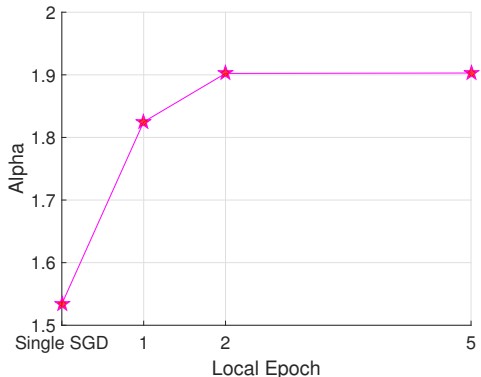

Figure 11: Estimation of $\alpha$ for Shakespeare dataset.

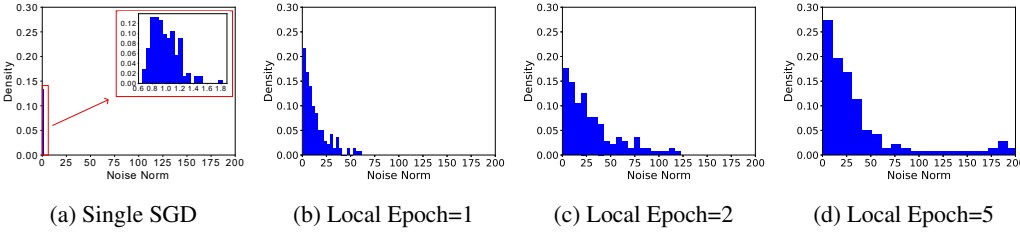

| (a) Single SGD | (b) Local Epoch=1 | (c) Local Epoch=2 | (d) Local Epoch=5 |

Figure 12: Distributions of the norms of the pseudo-gradient noises for Shakespeare dataset.

$m = 100$ clients participate in the experiment. The communication round is $T = 150$. The global learning rate is set to $1.0$, and the local learning rate is set to $0.1$. The batch size is chosen to be $64$.

### B.3.2 Experimental results

Table 3 shows the error rate of $\alpha$-value estimation in different cases, and this implies that the estimation of $\alpha$-value is within an acceptable margin of error.

## C Experiments in Section 5

In this section, we describe the details of the numerical experiments from Section 5 and provide some extra experimental results.

### C.1 Experiment details

### C.1.1 Strongly Convex Model with Synthetic Data

In these experiments, we consider a strongly convex model for Problem (1) as follows:

$$f_i(x) = \mathbb{E}_\xi\left[f_i(x, \xi)\right]$$

$$f_i(x, \xi) = \frac{1}{2}\|x\|^2 + \langle \xi, x \rangle,$$

where $x \in \mathbb{R}^{3 \times 1}$ and $\xi$ is a random vector. The optimal solution is $f(x^*) = 0$ with $x^* = [0; 0; 0]$.

Table 3: Error rate (%) of $\alpha$-value estimation.

|  | NonIID Index (p) | | | |
|---|---|---|---|---|
|  | 1 | 2 | 5 | 10 |
| Single SGD | -2.82 | -1.09 | -0.12 | 3.12 |
| Local Epoch=1 | 1.19 | 0.37 | 1.4 | 2.08 |
| Local Epoch=2 | 1.8 | 1.4 | 1.43 | 1.74 |
| Local Epoch=5 | 1.86 | 0.23 | 0.56 | 0.25 |

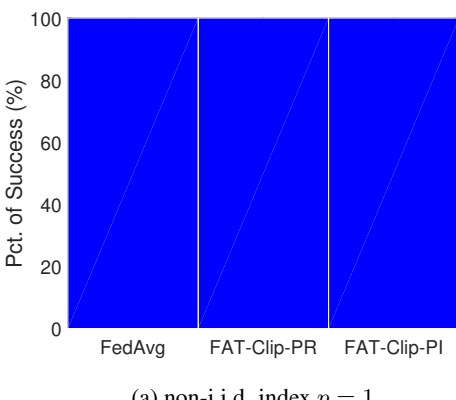

(a) non-i.i.d. index $p = 1$

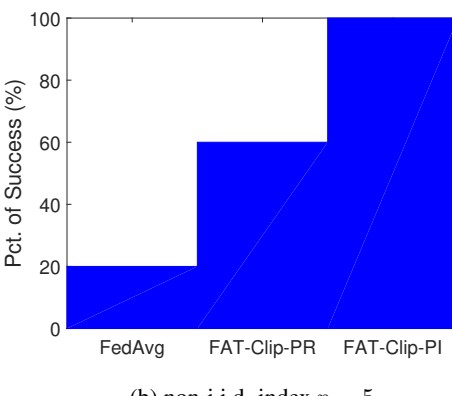

(b) non-i.i.d. index $p = 5$

Figure 13: Percentage of successful training over 5 trials when applying FedAvg, FAT-Clipping-PR and FAT-Clipping-PI to CIFAR-10 dataset in non-i.i.d. cases.

To compare the performance of FedAvg, FAT-Clipping-PI and FAT-Clipping-PR , we consider the noise $\xi$ to be a Cauchy distribution($\alpha < 2$, fat-tailed) with a location parameter of $0$ and a scale parameter of $2.1$.

To compare the performance of FAT-Clipping-PI and FAT-Clipping-PR under different scenarios, we consider the noise $\xi$ having different tail-indexes ($\alpha = 0.5, 1.0,$ and $1.5$) with the same location parameters of $0$ and the same scale parameters of $1$.

For all the distributions of $\xi$ mentioned above, we use the same experimental setup. There are $m = 5$ clients participating in the training. We choose the starting point $x_0 = [2; 1; 1.5]$. We set the global learning rate $\frac{\eta \eta_L}{m} = 0.1$ and the local learning rate $\eta_L = 0.1$. The local steps we use is $K = 2$, and the communication round is $T = 300$. The clipping parameter in FAT-Clipping-PI we select is $\lambda = 3$, and the clipping parameter in FAT-Clipping-PR is $\lambda = 5$.

### C.1.2 CNN (Non-convex Model) on the CIFAR-10

To test the performance of FAT-Clipping-PI and FAT-Clipping-PR for non-convex function, we run a convolutional neural network (CNN) on CIFAR-10 dataset. We compare FAT-Clipping-PI and FAT-Clipping-PR with FedAvg under different data heterogeneity.

In this experimental setting, we randomly select five clients from $m = 10$ clients to participate in each round of the training. The local epoch we use is two. The clipping parameter in FAT-Clipping-PI we select is $\lambda = 50$, and the clipping parameter in FAT-Clipping-PR is $\lambda = 2$. All the remaining settings are the same as described in Appendix B.1.1.

### C.2 Additional experimental Results

We provide two additional results when applying FedAvg, FAT-Clipping-PI and FAT-Clipping-PR to the CNN model on CIFAR-10 dataset. In Fig. 13, we show the percentage of successful training over 5 trials in non-i.i.d. cases when the non-i.i.d. index $p = 1$ and $p = 5$. These results further support our finding that FAT-Clipping methods and especially FAT-Clipping-PI reduce catastrophic training failures compared to FedAvg.