# OpenReview forum: "Taming Fat-Tailed (“Heavier-Tailed” with Potentially Infinite Variance) Noise in Federated Learning"
_NeurIPS.cc/2022/Conference — NeurIPS 2022 Accept_

### Official Review · Reviewer_cdKo · 2022-06-29

**Rating:** 5
**Confidence:** 4
**Soundness:** 4 excellent
**Presentation:** 3 good
**Contribution:** 2 fair

**Summary:**

This paper considers federated learning with stochastic gradient-based algorithms being run on each local machine, where the gradients may be potentially heavy-tailed in the strong sense that only a bounded $\alpha \in (1, 2]$ moment assumption is placed on the random gradient norms. Since traditional analysis of SGD makes use of bounded second moments, the authors consider the general technique of norm-clipped gradients, with two variants, namely clipping at each local iteration (per-iteration, PI variant), and clipping just before integrating the parameters from all local machines to the central server (per-round, PR variant).

The main contributions are theoretical. Their main results are in the form of excess error bounds in the strongly convex case, and approximate stationarity guarantees in the non-convex case, for both the PR and PI variants. As it turns out, clipping at each local iteration turns out to be the "better" strategy in terms of overall dependence on the number of samples used, an intuitive result, since without controlling the gradients at each iteration in the local processes (even with finite variance), we can say very little about what we'll get from the $K$ non-clipped local updates, and thus given the resolution of the standard analytical tools, such updates are essentially wasted. The authors share some lower bounds which give some perspective on these results, and complement their theory with some empirical tests that highlight the impact of the gradient noise on the convergence rate of each procedure. The overall story has "FedAvg" as the standard algorithm against which performance guarantees are compared, and the final message is that the PI variant is the natural robustification of FedAvg, since it recovers the rates achieved by FedAvg in the $\alpha = 2$ case.

**Questions:**

Please consider the points I raised in the previous field; if I have misconstrued the scenario considered in this paper, or misread any results, please let me know.

**Update:** the authors have clarified some concerns I had, and I have revised my score.

**Limitations:**

Overall, the authors are clear about what they intend to prove, what they do prove, and how the learning procedures tend to be have in practice.

**Strengths And Weaknesses:**

Overall, the paper is well-written, the main assumptions, learning algorithm details, and key results are stated clearly, and given a concise but transparent discussion. The empirical test results, while basic, do provide a very clear example highlighting how the degree of noise makes a big difference in the PR case, whereas the PI variant is less sensitive.

On the other hand, I had some concerns about how this work compares with some of the past literature on robust stochastic gradient methods. For example:

> High-probability Bounds for Non-Convex Stochastic Optimization with Heavy Tails, Ashok Cutkosky and Harsh Mehta, NeurIPS 2021

In this previous work, the setting is "centralized" (i.e., just one machine, rather than $m >1$ clients), but the authors obtain results for the non-convex, smooth objective case with $1 < \alpha \leq 2$ moment bounds on the stochastic gradients. These results hold with high probability, and can be shown to hold for the last iterate. The algorithm used takes advantage of clipping, normalizing, and momentum. As such, the state of the art for the centralized, smooth, non-convex, fat-tailed gradient learning problem has high-probability guarantees without random stopping. In comparison, the current work has results in expectation with random stopping (or the unknown best candidate). There seems to be a gap here, and I wonder if it is an unnecessary gap?

For example, set $K=1$, run the C and M (2021) sub-routine to process the noisy gradients from each local machine, and then average over these $m$ robust gradient estimators and update the central server parameters as in the current FAT-Clipping-* procedures. Or, alternatively, average over the non-clipped local gradients and then run the C and M robustification procedure before updating the central parameters. This latter case seems like it is just the C and M procedure with a mini-batch, and one would expect this to yield high-probability guarantees (I have not checked this carefully).

Of course, I know that in the federated learning setting we want the local machines to do the bulk of the work for us, and thus we'd like to take both $m > 1$ and $K > 1$ large; can we not modify the above robust procedure with $K=1$ to deal with the $K>1$ setting? If not, why not? The interplay between algorithmic decisions and the guarantees we can obtain are precisely the point of interest when moving from centralized to federated learning, and yet I feel these questions are sort of swept under the rug in the current analysis.

By the way, it is unclear to the reader if the error bounds in the convex case (e.g. Thm 1) are in expectation or are high-probability.

To summarize, it seems like the current paper is somewhat narrow-sighted in terms of the aggregation procedures (and the local procedures, for that matter) being considered, and state-of-the-art guarantees in the one-machine setting, which makes it somewhat difficult to evaluate the significance of this current work.

---

> ### Author Response · Authors · 2022-08-02
> **Response to Reviewer cdKo [1/2]**
>
> Thank you very much for your review and constructive comments, which helped us significantly improve the quality of this paper. In this revision, we have carefully revised our paper based on your comments, questions, and suggestions. The detailed point-by-point responses are as follows:
>
> > 1. Concerns about how this work compares with some of the past literature on robust stochastic gradient methods. For example:
> High-probability Bounds for Non-Convex Stochastic Optimization with Heavy Tails, Ashok Cutkosky and Harsh Mehta, NeurIPS 2021.
> In this previous work, the setting is "centralized" (i.e., just one machine, rather than $m > 1$  clients),  ...  In comparison, the current work has results in expectation with random stopping (or the unknown best candidate). There seems to be a gap here, and I wonder if it is an unnecessary gap?
> For example, set $K=1$, run the C and M (2021) sub-routine to process the noisy gradients from each local machine, and then average over these m robust gradient estimators and update the central server parameters as in the current $FAT-Clipping-*$ procedures. Or, alternatively, average over the non-clipped local gradients and then run the C and M robustification procedure before updating the central parameters. This latter case seems like it is just the C and M procedure with a mini-batch, and one would expect this to yield high-probability guarantees (I have not checked this carefully).
>
> **Our response:**  Thanks for your comments. Your understanding is correct that if we set local step $K=1$ for federated learning (FL) with full client participation, then FL is the same as mini-batch SGD. So, it is possible to extend the results of C and M (2021) from centralized learning to FL in such settings ($K=1$) and the same high-probability bound is expected to hold. However, this idea overlooks the core challenges introduced by FL, namely, local models’ discrepancy induced by data heterogeneity coupled with local steps ($K > 1$) for communication efficiency. It is non-trivial to understand the complex relationships between local update (i.e., $K$) and batch-size in FL (see [R1,R2]), let alone arguing the mini-batch SGD version of FL is more preferable (even if this version admits a high-probability guarantees). Also, we would like to emphasize on two aspects: 1) the community has spent a significant amount of efforts on in-expectation bounds (e.g., [2 - 10]) for FL under sub-Gaussian noise or noises with bounded variance assumption, and our extension of these results and understandings of tight convergence rates to heavy-tailed noise settings is still a major contribution; and 2) it is challenging to apply the procedure in C and M (2021) to FL setting ($K > 1$) and obtain the similar high-probability bound, please see our response to your Comment 2 for details.
>
> On the other hand, your understanding is correct that there is a gap in the sense that the state-of-the-art centralized SGD analysis could have high-probability bound and holds for the last iterate, while the current FL analyses (including our paper) can only provide an in-expectation bound. We believe there are three reasons: 1) studying centralized SGD has a longer history and thus more are known about SGD. In fact, the history of results on SGD started from in-expectation bounds and evolved to high-probability and last iterate bounds in recent years; 2) FL needs to handle data heterogeneity that is further complicated by local steps at clients. Thus, techniques showing these high-probability and last-iterate results for SGD cannot be immediately applied in FL; and 3) FL under heavy/fat-tailed noise is less explored, we believe it is completely acceptable that the current progresses remain focused on in-expectation bounds under heterogeneity challenge. To our knowledge, we are not aware of any high-probability bound for FL, even for Gaussian noise and let alone heavy/fat-tailed noise. For the question whether such a gap is necessary or not,  this is an excellent open question. We believe that the key in answering this question lies in understanding whether model discrepancy and ``client drift'' induced by the heterogeneity and local steps would deteriorate the probability guarantee in the learning process. In fact, we believe this is a promising topic and we are looking into it now. But this is beyond the scope of our paper now, and we thank you for raising this point!
>
> [R1] Blake Woodworth, Kumar Kshitij Patel, Sebastian Stich, Zhen Dai, Brian Bullins, Brendan Mcmahan, Ohad Shamir, Nathan Srebro, "Is Local SGD Better than Minibatch SGD?" ICML 2020.
>
> [R2] Prashant Khanduri, Pranay Sharma, Haibo Yang, Mingyi Hong, Jia Liu, Ketan Rajawat, and Pramod Varshney, "STEM: A Stochastic Two-Sided Momentum Algorithm Achieving Near-Optimal Sample and Communication Complexities for Federated Learning," NeurIPS 2021.

---

> > ### Author Response · Authors · 2022-08-02
> > **Response to Reviewer cdKo [2/2]**
> >
> > > 2. Of course, I know that in the federated learning setting we want the local machines to do the bulk of the work for us, and thus we'd like to take both $M > 1$ and $K > 1$ large; can we not modify the above robust procedure with $K = 1$ to deal with the $K > 1$ setting? If not, why not? The interplay between algorithmic decisions and the guarantees we can obtain are precisely the point of interest when moving from centralized to federated learning, and yet I feel these questions are sort of swept under the rug in the current analysis.
> >
> > **Our response:** Thanks for your comments. We believe the answer is ``no,'' in the sense that it is at least not a immediate extension from $K = 1$ to $K > 1$. Two key differences are as follows:
> >
> > 1) The procedure in C and M (2021) is actually a momentum-based SGD with extra parameter $\beta$ (or $\alpha$) and it scales with $T$. In other words, their results do not generalize to vanilla SGD with $\beta=0$. If one wants to directly employ such a momentum approach in FL, it could be problematic. Besides extra communication costs for transmitting momentum parameters, a more important question is that it is unclear how to construct the momentum in the local steps, since there is *no* synchronization among clients in local steps. On one hand, if we construct the momentum locally and separately for each client, the momentum used in each client would be different due to data heterogeneity. On the other hand, if an identical momentum specified by the server is used for each client, such momentum has a delay since momentum does not update with previous local steps. As a result, either a temporal or spatial discrepancy of the momentum may occur. Therefore, we do not see an immediate extension of momentum-based SGD in C and M (2021) for FL.
> >
> > 2) $K > 1$ would introduce the model discrepancy and client drifts among the client's local models due to data heterogeneity coupled with local steps. This is completely different from the case of $K = 1$. More specifically, the error term $\epsilon$ in C and M (2021) would be more challenging to bound. In C and M (2021), $\epsilon$ is bounded by an iterative equation (e.g., Eq (3) in their paper), which contains batch (stochastic) gradients based on the same position (denoted by $w$ in their paper). Utilizing the triangle inequality and $L$-smoothness assumption, the error term $\sum_{t = 1}^{T} \epsilon_t$ could be bounded when choosing proper learning rates and momentum hyper-parameters (Eq. (4) in their paper). However, for FL with $K > 1$, this is not the case. Following the same logic, $\epsilon$ would depend on batch (stochastic) gradients in different positions for FL due to local steps, i.e., $\sum_{i=1}^{m} \sum_{k =1}^{K} g(x_{t, i}^k$) (we use $x$ to denote the parameter in our paper). In such cases, we expect the model discrepancy ($x_{t, i}^k \forall i \in [m], k \in [K]$) would at least incur an extra constant term for the bound of $\epsilon$. As a result, the same results will not hold for FL with $K > 1$. Nonetheless, we completely agree that the interplay between algorithmic decisions and the guarantees we can obtain are excellent open questions when moving from centralized to federated learning. In centralized learning under heavy-tailed noise, we can already have tight bounds for clipping. When moving to FL, we show that clipping in each round (FAT-Clipping-PR) does not guarantee an optimal rate in terms of $K$, while clipping in each step (FAT-Clipping-PI) does yield an optimal convergence rate in $m$, $K$ and $T$. This shows the complex interplay between clipping and the convergence guarantees and more need to be investigated for heavy/fat-tailed noise in FL.
> >
> > > 3. By the way, it is unclear to the reader if the error bounds in the convex case (e.g. Thm 1) are in expectation or are high-probability.
> >
> > **Our response:** Thanks for the comments. They are in-expectation bounds and we will modify them correspondingly.

---

> > > ### Comment · Reviewer_cdKo · 2022-08-08
> > > **Re: Response to Reviewer cdKo**
> > >
> > > I thank the authors for their detailed response, and in particular the discussion of difficulties inherent in the federated learning scenario. I have revised my score to a borderline accept.

---

### Official Review · Reviewer_tQun · 2022-07-09

**Rating:** 6
**Confidence:** 3
**Soundness:** 3 good
**Presentation:** 3 good
**Contribution:** 2 fair

**Summary:**

The paper considers tackling fat-tailed noise in in federated learning, where the local gradients may not even have finite covariance. It is shown that in both theory and experiments, prior approaches can fail in this setting. The paper provides a simple solution to this problem by clipping the local updates. On the theory side, they show that the algorithms achieve near optimal convergence. The paper also provides numerical experiments to validate their results.

**Questions:**

It seems that the algorithms require knowledge of $\alpha$? (I noticed that $\lambda$ and $\eta$ settings in Theorem 1,2 both require $\alpha$) Is that an assumption that we can get rid of? Alternatively, what would be a good heuristic to set $\alpha$ in practice?

**Limitations:**

I do not see where the author(s) addressed the limitations of their work.

Federated learning itself may have potential negative societal impact, but I do not consider this work particularly risky in that regard. It's mostly theoretical.

**Strengths And Weaknesses:**

Strengths
----
The paper is generally well-written.

The main approach of the paper is simple and can be easily implemented in practice.

I have not checked the proofs in appendix, but the main theorem claims are strong, in the sense that they are optimal in $T,m$.

Weaknesses
---
It may be argued that the main algorithmic technique of this paper is not novel. It is mostly applying a known idea (clipping) to this problem.

Comparing the upper bound Theorem 1,2 and lower bound Theorem 5, the paper does not achieve optimal rate in terms of $K$, though the dependence on the more important factors of $T$ and $m$ is tight.

Minor
---
Line 252: constructing the lower fundamental bounds -> proving the following lower bounds

---

> ### Author Response · Authors · 2022-08-02
> **Response to Reviewer tQun**
>
> Thank you very much for your review and constructive comments, which helped us significantly improve the quality of this paper. In this revision, we have carefully revised our paper based on your comments, questions, and suggestions. The detailed point-by-point responses are as follows:
>
> > 1. It may be argued that the main algorithmic technique of this paper is not novel. It is mostly applying a known idea (clipping) to this problem.
>
> **Our response:** It is true that clipping is a known idea to tackle heavy-tailed noise. However, previous works are only focused on SGD in centralized learning. Due to the new challenges of data heterogeneity and the complications of local update steps in federated learning (FL) coupled with heavy-tailed noise, it remains *unclear* how one should develop a clipping-based algorithm for FL, which is exactly the motivation, novelty, and goal of this paper. To our knowledge, this is the first paper to investigate the design of convergent FL training algorithm under the heavy/fat-tailed noise in FL. The *new* and non-trivial question lies in whether the speedups of convergence rate in terms of local steps $K$ and clients $m$ are still achievable. In this paper, we answer this question affirmatively by proposing two clipping-based algorithms and showing their tight convergence rates.
>
> > 2. Comparing the upper bound Theorem 1,2 and lower bound Theorem 5, the paper does not achieve optimal rate in terms of K, though the dependence on the more important factors of  T and m is tight.
>
> **Our response:** Yes, you are right that the convergence rate of FAT-Clipping-PR does not match the optimal in terms of $K$. This provides fundamental understanding of clipping in FL and shows exactly that how to use clipping for FL under heavy-tailed noise is non-trivial, which motivates us to further develop FAT-Clipping-PI, which has tight convergence rate in terms of $m$, $K$ and $T$.
>
> > 3. It seems that the algorithms require knowledge of $\alpha$? (I noticed that $\lambda$ and $\eta$ settings in Theorem 1,2 both require $\alpha$) Is that an assumption that we can get rid of? Alternatively, what would be a good heuristic to set $\alpha$ in practice?
>
> **Our response:** Thanks for your constructive comments. It is true that, for our clipping methods to tackle heavy/fat-tailed noise, the choices of $\lambda$ and $\eta$ both depend on $\alpha$. The reason is that $\alpha$ represents the noise level. To ensure a proper update, the clipping and learning rates should be adjusted accordingly based on the noise, which means they should depend on $\alpha$. So we believe that such dependence can not be removed. Taking SGD with clipping as an example (Ref [22]), $\lambda$ and $\eta$ both depend on $\alpha$. In practice, our algorithm is not sensitive to the choices of $\lambda$ and $\eta$, which and are usually set as constants (see settings in appendix).
>
> We also want to point out that there is a typo in Theorems 2 and 4, and the dependence of $\lambda$ on $\alpha$ should be written as $\lambda = \Theta((mK^4T)^{\frac{1}{3\alpha-2}})$ and $\lambda = \Theta((mKT)^{\frac{1}{3\alpha-2}})$ for FAT-Clipping-PR and FAT-Clipping-PI, respectively. This means that, in practice, even though we may not have the knowledge of $\alpha$ in the beginning, this aforementioned ranges provide us sufficiently large room to set $\lambda$ appropriately. Further, as we observe more data during training, we will have better estimates of $\alpha$, so that we could set $\lambda$ more appropriately following the theoretical guidance in Theorems 2 and 4.
>
> > 4. Line 252: constructing the lower fundamental bounds -> proving the following lower bounds
>
> **Our response:** Thanks for pointing it out. We will correct it in the revision.

---

### Official Review · Reviewer_Kx2k · 2022-07-10

**Rating:** 7
**Confidence:** 4
**Soundness:** 4 excellent
**Presentation:** 4 excellent
**Contribution:** 4 excellent

**Summary:**

In this paper, the authors discussed the problems of the presence of fat-tailed noise in the stochastic gradient noise in the federated learning algorithm. The authors provided the empirical results that fat-tailed gradient noise can be induced by data heterogeneity and local update steps in the federated learning algorithm. And the fat-tailed gradient noise will lead to catastrophic training failures when applying GFedAvg algorithm. In order to fix that problem, the authors developed a clipping-based algorithmic framework called GAT-Clipping and provided the convergence rate bound of the algorithm.

**Questions:**

1) In the assumption the \alpha is defined between 1 and 2. But in the experiments of Figure 5, \alpha=0.5, which violates the assumption.
2) In both FAT-Clipping-PR and PI, there is an extra hyperparameter \lambda, and in the experiment part the value of \lambda is not mentioned. How to choose a proper \lambda value, is it related to the value of \alpha? If so, when we start the training, \alpha is not known, then how to choose \lambda.
3) The current experiments are not enough to validate the claim in the paper. Maybe providing extra experiments result can help the validation of the paper.

**Limitations:**

Yes, the authors adequately addressed the limitations and potential negative sociteal impact of their work.

**Strengths And Weaknesses:**

Strengths:
1) The paper is well-written and easy to follow the idea of the paper.
2) Related works have been fully discussed.
3) The idea is interesting and the proof of the theorem is solid.

Weakness:
1) The experiments are not enough to verify the claim.

---

> ### Author Response · Authors · 2022-08-02
> **Response to Reviewer Kx2k**
>
> Thank you very much for your review and constructive comments, which helped us significantly improve the quality of this paper. In this revision, we have carefully revised our paper based on your comments, questions, and suggestions. The detailed point-by-point responses are as follows:
>
> > 1. In the assumption the $\alpha$ is defined between 1 and 2. But in the experiments of Figure 5, $\alpha=0.5$, which violates the assumption.
>
> **Our response:** Yes, your observation is correct. In the theoretical analysis, we require $\alpha \in [1, 2]$, which is only a sufficient condition to have a well-defined expectation. But for experiments, we would like to push the limit of our algorithms and see whether $\alpha \in[1, 2]$ is also necessary. Interestingly, for synthetic data where we can control the noise level, we show that our proposed algorithms could even converge under the more severe Cauchy fat-tailed noise with $\alpha = 0.5$. We will clarify this in our revised version.
>
> > 2. In both FAT-Clipping-PR and PI, there is an extra hyperparameter $\lambda$, and in the experiment part the value of $\lambda$ is not mentioned. How to choose a proper $\lambda$ value, is it related to the value of $\alpha$? If so, when we start the training, $\alpha$ is not known, then how to choose $\lambda$.
>
> **Our response:** Thanks for the comments. For clipping-based methods, $\lambda$ is a standard hyper-parameter. In the theoretical analysis, $\lambda$ depends on $\alpha$ in the range $\lambda = \Theta((mK^4T)^{\frac{1}{3\alpha-2}})$ and $\lambda = \Theta((mKT)^{\frac{1}{3\alpha-2}})$ for FAT-Clipping-PR and FAT-Clipping-PI, respectively (see our Theorems 2 and 4). This means that, in the experiments, we can set $\lambda$ following these theoretical guidelines. Even though we may not have the knowledge of $\alpha$ at the beginning of training, the aforementioned ranges provide us large enough room to set $\lambda$ appropriately.
> In the experiments, we set $\lambda$ as a constant following an estimated range hinted by our theoretical guideline above for clipped SGD methods.
>
> > 3. The current experiments are not enough to validate the claim in the paper. Maybe providing extra experiments result can help the validation of the paper.
>
> **Our response:** Thanks for the comments. We do have more results in different settings of hyper-parameters such as learning rates and batch size, which are provided due to space and time constraints at the initial submission. But the main observations and conclusions remain the same in these extra experiments. We will add these results in the supplementary material in the revised version. In addition, we are running NLP tasks for the Shakespeare dataset and will add these results in our revised version.

---

### Official Review · Reviewer_ZT6s · 2022-07-12

**Rating:** 5
**Confidence:** 3
**Soundness:** 2 fair
**Presentation:** 1 poor
**Contribution:** 3 good

**Summary:**

The authors observe that federated learning setting creates gradients that are fat-tailed and that these schemes are often marred by divergence of the training. They propose two gradient clipping schemes that aim to mitigate this, the difference being where the gradient clipping is applied. The authors present convergence results for the two algorithms. The authors compare their results with an existing algorithm on synthetic and CIFAR-10 data.

**Questions:**

I presented my questions within my assessment of the paper above.

**Limitations:**

The authors present very little discussion of the weaknesses of their approach, which I think is not appropriate given the discussion presented above.

**Strengths And Weaknesses:**

Distributed learning schemes present crucial avenues of development for machine learning, especially with respect to topics such as privacy and scalability. Recent research on deep learning theory demonstrated the pervasive involvement of heavy-tailed distributed parameters and/or gradient noises in determining of various aspects of deep learning models such as convergence and generalization. Extension of these results to such crucial areas is an important endeavour. The most important contribution of the paper seems to the convergence rate results they present when their algorithms are used.

However I feel that the paper is insufficient in terms of both justifying the grounds for their methodology, as well as their effectiveness. This is also exacarbated by problems in the presentation which I will present below. The first problem is that the authors present insufficient evidence imprecisely when making a case for the existence of heavy tailed gradients in federated learning, and that these heavy-tailed gradient noise actually leads to divergence. The following questions remain unanswered or unclear in the text, which is problematic given the centrality of these matters for the article:

- How is pseudo-gradient noise calculated? How is the expected value for the pseudo-gradient computed during experiments?
- How does p relate to data heterogeneity exactly? How do we know it is a authoritative measure of data heterogeneity?
- Does Single SGD case refer to a normal training scheme in Fig. 2? If so, in what specific ways does a federated learning scheme differ from the known results of heavy-tailed noise in the literature?
- The authors use the divergence events as a motivation for gradient clipping. Is there a theoretical or empirical evidence to assume that the fat-tailed pseudogradient noise necessarily cause or even is statistically associated with divergence? The results on heavy-tailed parameter / gradient statistics in deep learning show that not only that this is not the case, but that heavy-tailed parameter / gradient statistics lead to better generalization.
- The authors analyze the pseudogradient noise and find it to be heavy-tailed. Do we know the tail index statistics of the expected-pseudogradient? (which can itself be heavy-tailed.)
So much so that it is not clear in the text distribution of which variable they are addressing. , the authors do not explicitly specify how they compute the pseudo-gradient noise (e.g. how do they compute the expected pseudo-gradient). The experiments in Fig. 1 and 2 seem

In addition to problems in the justification of the method, the authors do not demonstrate enough empirical evidence to show that their method is superior. The experiments are very small scale - the largest experiment setting only includes 5 trials for each setting. Given the weak evidence regarding the existence and/or the effects of heavy tailed gradient noise in federated learning, extensive empirical evidence can be expected to verify that the authors' methods are viable and advantageous. Especially important is the following question:

- Given the benevolent effects of heavy-tailed parameter statistics in deep learning, how do we know that the claimed advantages of the presented clipping schemes do not come with a negative effect on generalization performance?
- Given the importance of training hyperparameters in presence or absence of heavy-tailed parameter/gradient noise, how do the authors results change with respect to hyperparameters such as learning rate or batch size?

I would recommend a much more involved discussion of at least some of the questions presented above and as well as more extensive introduction to authors' terminology (e.g. the authors never explicitly state that x are their parameter vector.) The authors relegate some content to supplamentary material for lack of space: I believe a lot of space can be gained by not repeating Algorithms 2 and 3, and instead just describing the clipping operation in the text, referring to the original algorithm.

---

> ### Author Response · Authors · 2022-08-02
> **Response to Reviewer ZT6s [1/3]**
>
> Thank you very much for your review and constructive comments, which helped us significantly improve the quality of this paper. In this revision, we have carefully revised our paper based on your comments, questions, and suggestions. First of all, we would like to clarify one general misunderstanding about the main contributions of this paper. Our main contribution of this paper is to prove the tight convergence rates of federated learning (FL) under fat-tailed noise. To our knowledge, this is the first paper to consider FL under the fat-tailed noise setting. We have not focused on the generalization aspect of our proposed FL algorithms in the fat-tailed noise setting, and thus not claiming any contribution in generalization. But we have strong reasons to believe that our FL algorithms have good generalization performance, particularly in the over-parameterized deep learning regime (please see our detailed response below).
> The detailed point-by-point responses are as follows:
>
> > 1. How is pseudo-gradient noise calculated? How is the expected value for the pseudo-gradient computed during experiments?
>
> **Our response:** Thanks for your comments. We would like to clarify that the pseudo-gradient noise is calculated by faithfully following its mathematical definition. Specifically, the expected value of pseudo-gradient is evaluated by the averaging from all samples at all local datasets. Then, at each client, the pseudo-gradient noise is the difference between a stochastic gradient evaluated based on a mini-batch and expected value of pseudo-gradient. We believe that it is a similar process to analyze stochastic gradient noise for finite sum problems, e.g., in Ref. [15]. We will add the above clarifications in our revised version.
>
> > 2. How does p relate to data heterogeneity exactly? How do we know it is an authoritative measure of data heterogeneity?
>
> **Our response:** Thanks for your comment. As mentioned in the paper, our experiments are based on the MNIST (or CIFAR-10) dataset, which contains 10 label classes of digits. Hence, we follow a popular approach to generate non-i.i.d. datasets for FL based on MNIST: Here, we let $p$ represent the number of label classes in each client. As a result, $p=1$ represents the extreme non-i.i.d. case, while $p=10$ corresponding to the i.i.d. case. Therefore, $p$ can be viewed as an index for pathological non-i.i.d. partitions of the data. Note that this $p$-index for data heterogeneity has been widely used in many FL papers to experiment with non-i.i.d. data, including the seminal FedAvg paper with non-i.i.d. datasets in Ref. [1] and many others Refs. [2-10], and thus can be arguably viewed ``authoritative'' in certain sense.
>
> > 3. Does Single SGD case refer to a normal training scheme in Fig. 2? If so, in what specific ways does a federated learning scheme differ from the known results of heavy-tailed noise in the literature?
>
> **Our response:** Thanks for your question. `Single SGD' refers to FedAvg with a single local step ($K = 1$),  which is logically equivalent to mini-batch SGD. We will clarify this in our revised version. The goal of Fig. 2 is to illustrate how local steps could impact the $\alpha$-value, which characterizes the ``fatness'' of the tail.
> Federated learning with heavy/fat-tailed noise differs from traditional SGD with heavy-tailed noise in the following key aspects: i) federated learning features local update steps at each client, which does not exist in traditional mini-batch SGD-based training with heavy-tailed noise; ii) federated learning is typically associated with non-i.i.d. dataset (i.e., data heterogeneity) dispersed at each client -- a phenomenon that does not exist traditional mini-batch SGD-based training; and iii) as evidenced in our experiments, the severity of heavy/fat-tailed property is intimately linked with the choices of local update steps and the degree of data heterogeneity, which is also unseen in traditional mini-batch SGD-based training with heavy-tailed noise.

---

> > ### Author Response · Authors · 2022-08-02
> > **Response to Reviewer ZT6s [2/3]**
> >
> > > 4. The authors use the divergence events as a motivation for gradient clipping. Is there a theoretical or empirical evidence to assume that the fat-tailed pseudogradient noise necessarily cause or even is statistically associated with divergence? The results on heavy-tailed parameter / gradient statistics in deep learning show that not only that this is not the case, but that heavy-tailed parameter / gradient statistics lead to better generalization.
> >
> > **Our response:** Thanks for your comments. Since this comment contains two points, we structure our response into two parts accordingly.
> >
> > - *Fat-tailed Noise Leading to Divergence?* The answer to this question is ``Yes,'' and we have both theoretical and empirical evidence. For theoretical evidence, we have studied the impacts of fat-tailed noise on FL in Section 3 (Line 168-180). We can see that, under fat-tailed noise, the FedAvg algorithm is not guaranteed to converge in the sense that there exists a function for which the FedAvg algorithm diverges if one does not attempt to mitigate the fat-tailed noise. For empirical evidence, we observed from our experiments: 1) for synthetic data where we can fully control the noise's tail index (i.e., $\alpha$-value), FedAvg does not converge under fat-tailed noise (Fig. 3) but converges under Gaussian noise with the exact same settings; and 2) for real-world non-i.i.d. data, data heterogeneity leads to a more severe fat-tailed phenomenon (i.e., smaller $\alpha$-value) (Fig. 2), and thus being more likely to diverge (Fig. 7).
> >
> > - *Relationship with Generalization:* To our knowledge, we are unaware of any references that point out the non-convergence under heavy-tailed parameter/gradient statistics imply good generalization. We would highly appreciate if Reviewer ZT6s could provide pointers to such references, and we will be happy to cite them and provide a thorough discussion. On the other hand, we are aware of a prominent and interesting line of research called 'double descent' that links better convergence in training with good generalization, particularly in the *over-parameterized* (including deep learning) regime. The area is also referred to as 'benign overfitting' in the literature. Specifically, it has been theoretically shown that, in the over-parameterized regime, the so-called `Polyak-Lojasiewicz' (PL) condition (also known as gradient dominant condition) tends to hold with with high probability [R1, R2]. Thus,  *any stationary point* is a global minimum solution with a *zero loss*. Moreover, many works either empirically [R3] or theoretically (e.g., [R4-R6] and many others) showed that such a zero-loss overfitting in the over-parameterized regime enjoy good generalization performances through the lens of 'double descent.' To summarize, combining the above arguments suggests the following logical link:  *'PL condition happens in deep learning w.h.p.' plus 'Better convergence in training of over-parameterized models to reach zero-loss' $\Rightarrow$ Better generalization in deep learning.*
> >
> > [R1] Chaoyue Liu, Libin Zhu, Mikhail Belkin, "Loss landscapes and optimization in over-parameterized non-linear systems and neural networks," ACHA 2022.
> >
> > [R2] Chaoyue Liu, Libin Zhu, Mikhail Belkin, "On the linearity of large non-linear models: when and why the tangent kernel is constant," NeurIPS 2020.
> >
> > [R3] Mikhail Belkin, Daniel Hsu, Siyuan Ma, Soumik Mandal, "Reconciling modern machine learning practice and the bias-variance trade-off," PNAS, 2019, 116 (32).
> >
> > [R4] Peizhong Ju, Xiaojun Lin, Jia Liu, "Overfitting Can Be Harmless for Basis Pursuit, But Only to a Degree," in Proc. NeurIPS 2019.
> >
> > [R5] Song Mei and Andrea Montanari, "The generalization error of random features regression: Precise asymptotics and double descent curve," CPAM 2021.
> >
> > [R6] Mikhail Belkin, Daniel Hsu, Ji Xu, "Two models of double descent for weak features," SIAM Journal on Mathematics of Data Science, 2(4), 2020.
> >
> > > 5. The authors analyze the pseudo-gradient noise and find it to be heavy-tailed. Do we know the tail index statistics of the expected pseudo-gradient? (which can itself be heavy-tailed.) So much so that it is not clear in the text distribution of which variable they are addressing. , the authors do not explicitly specify how they compute the pseudo-gradient noise (e.g. how do they compute the expected pseudo-gradient).
> >
> > **Our response:** We believe there is misunderstanding here, and we would like to further clarify. We evaluate pseudo-gradient noise following a similar process for SGD noise. We calculate one expected-pseudo gradient based on all data samples from all clients (i.e., similar to calculating full gradient), as detailed in response to Comment 1. We will add these clarifications to the revised version. Thanks! We analyze the $\alpha$-index of the pseudo-gradient noise following the same procedure of that in SGD in Ref. [15]. For the computation of expected pseudo-gradient, see our response for Comment 2.

---

> > > ### Author Response · Authors · 2022-08-02
> > > **Response to Reviewer ZT6s [3/3]**
> > >
> > > > 6. Given the benevolent effects of heavy-tailed parameter statistics in deep learning, how do we know that the claimed advantages of the presented clipping schemes do not come with a negative effect on generalization performance? Maybe by clipping you are killing generalization.}
> > >
> > > **Our response:** Thanks for your comments. Generalization is a relevant but relatively independent topic. Our paper focuses on the convergence rate of FL training under heavy-tailed noise, which is the first study under such settings in FL, thus being novel compared to current convergence analyses under Gaussian noises or noises with bounded variance. Hence, our main contribution is on convergent FL algorithm design under fat-tailed noise and its convergence rate analysis, rather than its generalization performance. Again, in terms of generalization, we are not aware of any works showing that heavy-tailed parameter statistics imply better generalization bound in SGD-based algorithms. We would highly appreciate it if the reviewer could provide pointers to such references. Regarding the effect of clipping on generalization, clipping is widely-used in current deep learning for faster and more stable training, e.g., [17, 18]. Again, following the logical link in the response to your Comment 4, one has strong reasons to believe that better convergence assisted by clipping leads to better interpolation of training data in the over-parameterized/deep learning regime, which in turn implies good generalization due to the 'double descent' effect.
> > >
> > > > 7. Given the importance of training hyperparameters in the presence or absence of heavy-tailed parameter/gradient noise, how do the authors' results change with respect to hyperparameters such as learning rate or batch size?
> > >
> > > **Our response:** Thanks for the comments.  We note that there are many hyper-parameters (learning rate, batch size, local steps, communication round, etc.) that have an impact on algorithmic performance. To give fair comparisons between heavy-tailed noise and Gaussian noise, all hyper-parameters used in the two cases are exactly the same. We have results in different settings of learning rates and batch size, but they all show the same observations and conclusions. In addition, we extend our results by averaging over 10 trials. The new results show the same observations: the success rates of FAT-Clip-PR changes from $60\%$ to $40\%$ while others remain the same. We will add these results in the revised version.

---

> > > > ### Comment · Reviewer_ZT6s · 2022-08-09
> > > > **Thank you**
> > > >
> > > > I thank the authors for their comments, which helped me understand how they situate their work better. Re. authors' comment: "we are not aware of any works showing that heavy-tailed parameter statistics imply better generalization bound in SGD-based algorithms" please see the works below:
> > > >
> > > > - https://papers.nips.cc/paper/2020/hash/37693cfc748049e45d87b8c7d8b9aacd-Abstract.html
> > > > - https://arxiv.org/abs/1901.08278
> > > >
> > > > I raise my score to reflect authors' feedback.

---

> > > > > ### Author Response · Authors · 2022-08-09
> > > > > **Response to Reviewer ZT6s' Second Feedback**
> > > > >
> > > > > Thanks so much for raising your score! Also, thanks so much for the pointers to the references on heavy-tailed statistics and generalization! We will add discussions on these references you provided in our revised paper.
> > > > >
> > > > > Best,
> > > > > Authors

---

### Official Review · Reviewer_4F3m · 2022-09-02

**Rating:** 6
**Confidence:** 5
**Soundness:** 3 good
**Presentation:** 3 good
**Contribution:** 2 fair

**Summary:**

It has previously been shown that cross-device federated learning (FL) can exhibit unstable convergence and experiences a sudden catastrophic failure. The current work posits that this instability is due to fat-tailed nature of the client updates. They provide some empirical evidence for this hypothesis, and proceed to show how to combine clipping with FedAvg to derive a new algorithm which converges under such fat tailed noise.

**Questions:**

Since this is an emergency review for which the authors may not be able to respond in time, I will only use this space to summarize my review.

Overall, this work proposes a new hypothesis of why non-iid-ness is challenging in cross-device federated learning - they claim it is due to fat tailed noise. While there is some preliminary evidence for this, the results are far from conclusive. Some of the authors' own experiments contradict this and further fat-tailed-ness does not explain other empirical phenomenon observed. Finally, there are places where both the experiments and the theory can be improved.

Despite such shortcomings, the paper provides a valuable new perspective on FL and advances the conversation. I feel that these shortcomings are worth investigating by the community in future work. However, I **strongly recommend** that the authors to replicate their experiments on a few other datasets / models in the final version of their paper. I recommend acceptance trusting the authors to follow through on this.

**Limitations:**

No societal limitations.

**Strengths And Weaknesses:**

## Strengths

1. The problem of unstable convergence is an important aspect, and the hypothesis that this may be a result of fat tailed noise is mathematically sound. Such noise would indeed explain the catastrophic failure seen in federated learning.

2. I very much appreciate the experiments in Section 3 (Figs. 1, 2, and 3) to establish the presence of fat tailed noise in FL. The setup is carefully thought through and indeed shows that fat tailed noise increasingly becomes a problem as heterogeneity increases. This explains why we see unstable training in non-iid setting, but not in an iid setup.

3. The algorithms derived to deal with fat tailed noise FAT-CPR (where the client pseudo gradient is clipped), and FAT-CPI (where each local update is clipped) are both intuitive and shown to have strong convergence guarantees as well.

4. The theory shows that FAT-CPI has a faster convergence rate than FAT-CPR. This is exactly reflected in practice as well where the former seems to perform better. Overall, FAT-CPI is a simple, intuitive algorithm which is provably more stable under fat tailed noise.

## Weakness

The paper does still leave a lot of questions unanswered however.

1. The estimated alpha in Fig. 2 seems to indicate that increasing the number of local steps makes the distribution less fat tailed. However, we know from experience that more local steps makes the convergence more brittle. In fact, Fig 2 seems to indicate that standard single-SGD is very fat tailed, but empirically converges without any issues. This is not discussed in the text.

2. Previous work on large cohort training [Z. Charles et al. 2021] shows that the updates seem to become orthogonal as we increase number of local steps. How does that explanation fit in with the current fat-tailed explanation? More generally, there is no investigation of why such fat tailed distributions seem to arise in federated learning.

3. Assumption 3 makes no distinction between the local variance within a client vs. variance across the clients due to heterogeneity. Thus, the theory is not useful to shed light on the effects on non-iid ness. Further, unlike previous work of [J. Zhang et al. 2020], the current work does not investigate adaptivity.

4. Finally, the experimental results are very limited - only CIFAR 10 with a CNN is tried. It is unclear how this observation and algorithm would generalize to other settings.

---

### Official Review · Reviewer_Eovh · 2022-09-02

**Rating:** 4
**Confidence:** 4
**Soundness:** 3 good
**Presentation:** 3 good
**Contribution:** 2 fair

**Summary:**

(I have been added as a reviewer only very late in the review process. Even though I am adding a few questions below I am fully aware that the authors cannot respond appropriately. Nevertheless, I believe these remarks might still be useful for the internal discussion an AC).

This paper studies stochastic gradient descent in a distributed/federated learning setting under the assumption that the $\alpha$-th moment ($\alpha \in (1,2]$) is bounded. This generalizes standard analyzes, that typically assume bounded variance (second moment).

Earlier work [22] studied SGD in a centralized setting ($m=1$ worker) under the same bounded moment assumptions. This work showed that gradient clipping allows SGD to converge, and this earlier work also showed that this approach is optimal by proving matching lower and upper bound.

The paper under review generalizes the results from [22] to a distributed setting. The main results are:

(i) an asymptotic convergence analysis for two version of clipping (per-iteration or per-epoch) with local steps under the assumption that the functions on the $m$ workers are identical (homogenenous), for the strongly convex and non-convex settings. The convergence guarantee of one of the presented algorithms matches the lower bound presented in [22] and is thus optimal.

(ii) the paper makes the conjecture -- which I find very interesting -- that clipping can also be used to overcome non-iid-ness when the functions on the $m$ workers are different (without resorting to variance reduction techniques such as SCAFFOLD or similar). However, this claim is only verified in one numerical experiment and the presented results are thus not yet conclusive enough to prove this claim.


**Questions:**

- it is not so evident that $c$ is constant. (The expression that $c$ must satisfy depends on parameters like $T,K$, etc.). Please show here that $c$ is indeed constant.

- why does the paper introduce two algorithms (PR, PI), when apparently one is dominated by the other (at least in theory). Please make it more clear what would be the advantages of using the suboptimal algorithm in practice (or why it is included in the paper).

- to get the correct dependency on $K$ for the case $\alpha = 2$ a separate proof is needed (Theorems 6&7). One could expect to recover the result by letting $\alpha \to 2$, which is not the case. Is this an artefact of the proof, or is there a deeper reason that the case $\alpha < 2$ is behaving differently?

- how is the parameter $\alpha$ estimated in Figure 2? Please explain the method.

**Limitations:**

Already mentioned above that limitations of theoretical results, and practical guidelines for choosing PR vs. PI could have been emphasized better.

**Strengths And Weaknesses:**

The paper is well written and easy to follow. In terms of originality, it seems the paper combines the clipping technique from [22] with other known analyzes for local SGD/fedAvg. It would have been nice if the paper could have pointed out the technical difficulties a bit more to explain to the reader the significance of the contribution regarding these aspects.

As mentioned above, I particular liked the discussion of the general heterogeneous setting as this seems to cover some novel aspects. Unfortunately, the evaluation fell a bit short.

The convergence results are presented from an asymptotic point of view and only hold for sufficiently large iteration counter $T$. I tried to estimate a lower bound on $T$: by the first two conditions stated in Theorem 1 on the term $\eta \eta_L K$, it must hold $\frac{c \ln T}{mK} \geq 1$, i.e. $T \geq e^{mK}$ when ignoring $c$. An exponential dependence on $m$ and $K$ seems very strong here, as the asymptotic regime might never be reached in practical settings where $m$ and $K$ are typically large. I was missing a discussion these limitations of the theoretical result. This strong dependence makes me doubt the practical relevance of the current theory (and might also hint that these results could probably be improved).

Calling Theorem 5 a theorem seems a bit overclaimed, as the result is a direct corollary of [22].

A theoretical analysis of the heterogeneous setting would strengthen the paper considerably and would allow the contribution to be clearly distinguished from previous work.

---

### Author Response · Authors · 2022-08-02
**General Response**

We thank all reviewers for your constructive and insightful comments! As a general response, we would like to highlight the main contribution of our paper. That is, the tight in-expectation bounds in federated learning (FL) under heavy-tailed stochastic gradients, showing the optimal rates in $m$, $K$ and $T$. We note that some other topics are mentioned by reviewers, such as generalization, high-probability bound, and last iterate guarantees. We believe these are important and promising research topics, but they are beyond the scope of our current paper.

---

### Meta-Review · Area_Chair_aTTp · 2022-08-23

**Recommendation:** Accept
**Confidence:** Certain

**Metareview:**

This is a borderline paper, and we ended up soliciting two additional reviewers (4F3m and Eovh) for additional feedback after the main review/author discussion period.

The general sentiment shared by the reviewers (including the two additional ones) is that the paper studies an interesting/important problem and provides some interesting discussion and results -- albeit largely based on natural extensions of existing methods (e.g. in the centralized setting). Some of these results could also be improved (both in terms of better discussion on novelty and just stronger guarantees), but the reviewers generally appreciate that the authors have started discussion and work in this area.

Multiple reviewers think that the hypothesis relating non-iidness with fat tailed distributions is interesting, but are underwhelmed by the supporting analysis and commentary. It would be great if the authors could improve this discussion, either with more concrete analysis or with better empirical evidence.

The biggest criticism of this paper that is shared by most if not all of the reviewers is that the current experimental section is severely lacking. The authors mentioned that they would add in more experiments in the final draft of the paper (e.g. https://openreview.net/forum?id=8SilFGuXgmk&noteId=ShuBB-7e3B-). This response seemed satisfactory to the reviewers, so this paper will be accepted based on the premise that the authors will follow up on that promise. In particular, it is expected that the authors will go over the reviews and try to carefully address all comments about the experiments (e.g. replicating experiments on at least a few other datasets and models -- ideally at larger scale, using experiments to better verify the non-iid hypothesis, designing experiments that verify the theoretical guarantees).

**Award:**

No

---

### Decision · Program_Chairs · 2022-09-14

Accept